# MeMoSORT: Memory-Assisted Filtering and Motion-Adaptive Association Metric for Multi-Person Tracking

## Abstract

Multi-object tracking (MOT) in human-dominant scenarios, which involves continuously tracking multiple people within video sequences, remains a significant challenge in computer vision due to targets' complex motion and severe occlusions. Conventional tracking-by-detection methods are fundamentally limited by their reliance on Kalman filter (KF) and rigid Intersection over Union (IoU)-based association. The motion model in KF often mismatches real-world object dynamics, causing filtering errors, while rigid association struggles under occlusions, leading to identity switches or target loss. To address these issues, we propose MeMoSORT, a simple, online, and real-time MOT algorithm with two key innovations. At first, the Memory-assisted Kalman filter (MeKF) uses memory-augmented neural networks to compensate for mismatches between assumed and actual object motion. Secondly, the Motion-adaptive IoU (Mo-IoU) adaptively expands the matching region and incorporates height similarity to reduce mis-associations, while remaining lightweight. Experiments show that MeMoSORT achieves state-of-the-art performance, with HOTA scores of 67.9% and 82.1% on DanceTrack and SportsMOT, respectively.

## 1 Introduction

Multi-object tracking (MOT) refers to the task of continuously tracking multiple objects across video sequences, and has been widely applied in autonomous driving (Geiger et al., 2012; Yu et al., 2020), video surveillance (Milan et al., 2016; Dendorfer et al., 2020), and sports analysis (Cui et al., 2023; Cioppa et al., 2022; Sun et al., 2022). Among these scenarios, tracking persons has become the most extensively studied and practically relevant subproblem.

As the dominant paradigm of MOT, tracking-by-detection (TBD) (Bewley et al., 2016; Zhang et al., 2022; Cao et al., 2023; Maggiolino et al., 2023) addresses this task by decomposing it into three key stages: detection, state estimation (filter), and association. While detection accuracy was historically a primary limiting factor, the advent of high-performance detectors like YOLO series (Redmon et al., 2016; Varghese & M., 2024) has largely addressed this issue. As a result, the performance of modern TBD trackers is now principally constrained by the efficacy of the other two stages: state estimation and association.

Conventional state estimation and association modules suffer from two key limitations. First, the Kalman filter (KF) (Kalman, 1960) assumes linear dynamics and a first-order Markovian process (Khodarahmi & Maihami, 2023), which does not match the complex and temporally correlated motion patterns of real-world targets (as illustrated in Appendix A). The mismatch can lead to significant errors in motion prediction and estimation when the actual motion deviates from these assumptions (Wang, 2025), such as in coordinated or repetitive behaviors (e.g., a dancer consistently spinning after a specific jump). Second, standard association strategies often rely on simplistic Intersection over Union (IoU) (Yu et al., 2016), without adapting to the target's motion patterns. This lack of adaptability can degrade association performance, resulting in tracking failure.

To address these challenges, we propose MeMoSORT, a simple, online, and real-time MOT framework tailored for complex scenarios. MeMoSORT introduces two key innovations: **(a)** Memory-assisted Kalman Filter (MeKF), which leverages memory-augmented neural networks (NN) to com-

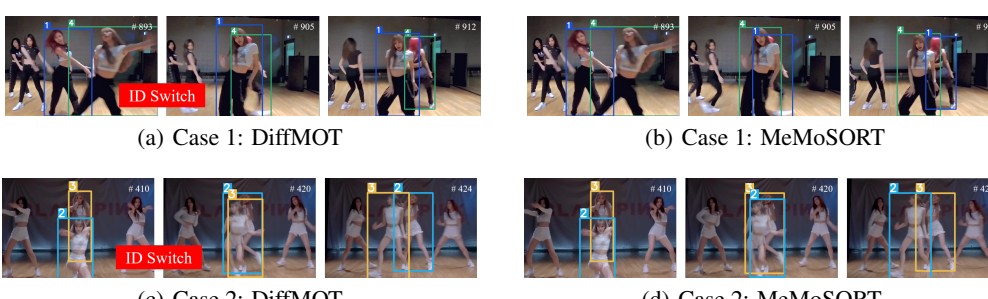

(a) Case 1: DiffMOT

(b) Case 1: MeMoSORT

(c) Case 2: DiffMOT

(d) Case 2: MeMoSORT

Figure 1: Visualization of DiffMOT (a, c) and MeMoSORT (b, d) in challenging scenarios from the DanceTrack validation set. **Case 1 (Complex Motion)**: DiffMOT's inaccurate prediction leads to an identity switch, while MeMoSORT maintains the correct identity by leveraging the precise state estimation from its MeKF. **Case 2 (Severe Occlusion)**: Standard IoU-based association in DiffMOT fail in association when encountering severe occlusion. MeMoSORT's Mo-IoU robustly handles this challenge and ensuring continuous tracking.

pensate for the gap between assumed and actual motion patterns; **(b)** Motion-adaptive IoU (Mo-IoU), which adaptively expands the matching region and incorporates height similarity to reduce association errors.

Extensive experiments demonstrate that MeMoSORT achieves state-of-the-art (SOTA) performance on challenging benchmarks, reaching HOTA scores of 67.9% on DanceTrack and 82.1% on SportsMOT, significantly outperforming existing methods across multiple metrics.

## 2 RELATED WORKS

### 2.1 METHODS FOR STATE ESTIMATION

KF is the widely used for state estimation in early TBD trackers. Subsequent methods such as OC-SORT (Cao et al., 2023) introduced improvements to handle occlusions, but could not overcome the fundamental limitations of the linear, first-order Markovian motion model in scenarios with complex, non-Markovian dynamics.

To address this, one line of research replaces the KF entirely with data-driven NN. For example, Diff-MOT (Lv et al., 2024) employs a diffusion model for non-linear motion prediction, while Mamba-based trackers (Xiao et al., 2024a; Khanna et al., 2025) utilize state space models to capture complex motion. However, a key challenge for these pure predictors is the lack of a principled filtering step; they often replace a track's state directly with the noisy detector measurement instead of update, which degrade trajectory quality.

Another direction (Li et al., 2024; Adžemović et al., 2025) involves hybrid approaches that replace physics-based models with deep learning techniques within the classic Bayesian filter structure. These methods combine the expressiveness of NN with the stability of the prediction–update cycle. A drawback is that discarding the physics-based prior in favor of a complex NN makes the filter heavily reliant on training data, thereby reducing robustness and generalization.

### 2.2 ASSOCIATION BETWEEN DETECTION AND PREDICTION

Mainstream association methods within the TBD paradigm are typically based on two principles: spatial consistency and appearance similarity. The former is primarily addressed by IoU and its variants, while the latter relies mainly on ReID based methods. In practice, these two approaches are often combined into a final association cost, typically through a weighted sum.

IoU-based methods use IoU as spatial association metric, higher IoU between boxes across frames represents higher probability of the same targets. Recent studies modified IoU by expanding the scale of the box (Fan et al., 2023; Huang et al., 2024b), incorporating height similarity (Yang et al.,

2024) or considering both (Khanna et al., 2025). However, the performance of above types of IoU with fixed parameters critically depends on manual setting, limiting their applicability across complex environments. Existing dynamic parameter methods either use multiple association stages with several fixed parameter (Huang et al., 2024b) or focus on temporal information of the trajectory (Stanojević & Todorović, 2024), lacking adaptivity according to target's motion characteristics.

ReID-based methods uses an additional NN to extract feature to represent the visual appearance of target, considering shorter distance between feature across frames leads to same target. The majority of ReID based methods (Wojke et al., 2017; Aharon et al., 2022; Du et al., 2023) use convolution NN to extract appearance feature and apply cosine distance as measurement. ReID-based methods are less effective in distinguishing targets with similar appearance or under occlusion.

## 3 METHODOLOGY

### 3.1 PRELIMINARIES: TRACKING BY DETECTION

The TBD paradigm is a prevalent approach in MOT. Unlike monolithic end-to-end methods, TBD frameworks decouple the tracking problem into three distinct stages, as illustrated in Figure 2(a): detection, association, and filtering.

The first step involves an object detector, such as the widely used YOLO model, generating a set of candidate boxes for each frame $t$. A detection is typically represented as a vector $\widetilde{\boldsymbol{b}}_t = [\widetilde{x}_t, \widetilde{y}_t, \widetilde{w}_t, \widetilde{h}_t]^\top$, defining the center coordinates, width, and height of the box. It is generated via the linear measurement matrix $\mathbf{H}$ from the target's state vector, $\boldsymbol{b}_t$, which contains the target's position, size, and velocity. This relationship is modeled as:

$$\widetilde{\boldsymbol{b}}_t = \mathbf{H}\boldsymbol{b}_t + \boldsymbol{v}_t, \tag{1}$$

where $\boldsymbol{v}_t$ is the measurement noise, it is generally assumed to follow an independent zero-mean Gaussian distribution with a covariance matrix $\mathbf{R}_t$.

The output detections, which are prone to false alarms and misses from occlusion, are linked across frames via association to form trajectories. This association is formulated as a bipartite matching problem between existing tracks and current detections, where the matching cost typically combines spatial overlap (IoU) and appearance similarity (ReID). Specifically, IoU measures the spatial overlap between a detection $\widetilde{\boldsymbol{b}}_t$ and a track's predicted state $\hat{\boldsymbol{b}}'_t$. And ReID involves masking the object within the detection box, encoding its appearance, and then measuring similarity using cosine distance. Finally, the Hungarian algorithm is used to find the optimal assignments based on the combined matching cost.

After association, a filter is applied to estimate the target's state via a prediction-update cycle. For the widely used KF, the prediction is based on a linear, first-order Markovian motion model:

$$\boldsymbol{b}_t = \mathbf{F}\boldsymbol{b}_{t-1} + \boldsymbol{w}_t, \tag{2}$$

where $\mathbf{F}$ is the linear state transition matrix (e.g. constant velocity model). And $\boldsymbol{w_t}$ is the process noise, it is generally assumed to follow an independent zero-mean Gaussian distribution with a covariance matrix $\mathbf{Q}_t$. In the update step, this prediction is refined by incorporating the newly associated detection.

However, this prevalent pipeline suffers from two critical limitations. First, the state estimation relies on an underlying linear, first-order Markovian motion model is often an oversimplification of real-world dynamics. This prevents the KF from handling complex, non-linear paths. Second, the association cost, based mainly on IoU, is unreliable during occlusion as the boxes is mixed to a mess. To this end, our work introduces a deep learning aided filter that leverages temporal memory to model complex dynamics and a robust association metric resilient to occlusion.

### 3.2 FRAMEWORK OF THE PROPOSED MEMOSORT

The framework of our proposed MeMoSORT is illustrated in Figure 2(b), with the following three stages.

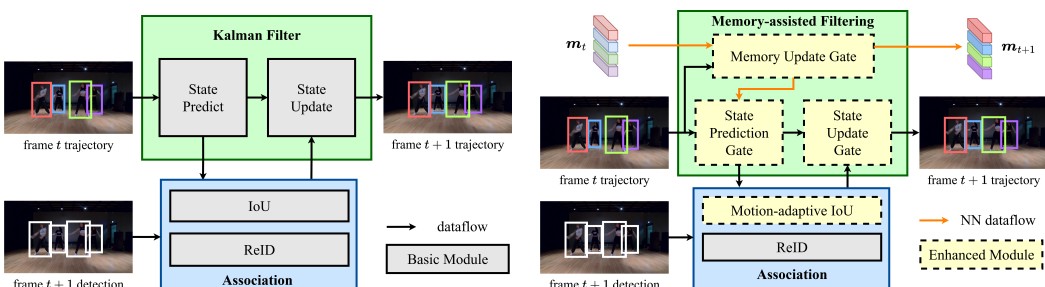

(a) Framework of Tracking-by-Detection      (b) Framework of MeMoSORT

Figure 2: Comparison between (a) the conventional Tracking-by-Detection framework and (b) our proposed MeMoSORT framework. MeMoSORT introduces two key components: it leverages a memory mechanism to guide state estimation for more accurate state prediction and update, and it applies a Motion-adaptive IoU to achieve robust association.

**Detection.** In line with the conventional TBD paradigm, MeMoSORT leverages the YOLOX (Ge et al., 2021) to perform the initial detection task, generating a set of candidate boxes for all potential targets within each frame.

**Association.** We introduce an association pipeline inspired by Deep OC-SORT (Maggiolino et al., 2023). This pipeline incorporates our novel Mo-IoU, a metric that refines conventional IoU by adaptively expanding the boxes and considering height similarity based on the target's motion characteristics. Within this pipeline, detections are initially stratified by their confidence scores. High-scoring detections are matched using a combined Mo-IoU and ReID cost via the Hungarian algorithm, while low-scoring detections are matched using a standard IoU cost.

**Filtering.** We propose the MeKF, a variant of the standard KF inspired by literature (Yan et al., 2024) that leverages memory to aid in state estimation. The MeKF consists of three gated modules: a Memory Update Gate (MUG) to maintain a historical representation, a State Prediction Gate (SPG) to correct the motion prediction using memory, and a State Update Gate (SUG) to refine the state based on the associated detection.

### 3.3 MEMORY-ASSISTED KALMAN FILTER

To address the limitations of the first-order Markovian assumption in the KF (Eq. 2), we introduce a non-Markovian motion formulation capable of modeling the complex dynamics inherent in real-world targets:

$$\boldsymbol{b}_t = f_t(\boldsymbol{b}_{t-1}, \boldsymbol{b}_{t-2}, ..., \boldsymbol{b}_1) + \boldsymbol{w}_t, \tag{3}$$

where $f_t(\cdot)$ is a non-linear transition function. Unlike the transition matrix $\mathbf{F}$ in Eq. 2, $f_t(\cdot)$ explicitly conditions the state prediction on the full trajectory history, thus enabling the modeling of long-term dependencies. As an explicit analytical form for $f_t(\cdot)$ is intractable, we simplified the problem by introducing the transition matrix $\mathbf{F}$, namely,

$$\boldsymbol{b}_t = \mathbf{F}\boldsymbol{b}_{t-1} + \underbrace{f_t(\boldsymbol{b}_{t-1}, \boldsymbol{b}_{t-2}, ..., \boldsymbol{b}_1) - \mathbf{F}\boldsymbol{b}_{t-1}}_{\boldsymbol{\Delta}_t^{\mathbb{F}}} + \boldsymbol{w}_t$$

$$= \mathbf{F}\boldsymbol{b}_{t-1} + \boldsymbol{\Delta}_t^{\mathbb{F}} + \boldsymbol{w}_t, \tag{4}$$

where $\boldsymbol{\Delta}_t^{\mathbb{F}}$ is the model mismatch term, capturing the residual between the non-Markovian and first-order Markovian dynamics. As this term is a function of the entire history, we approximate it using a mapping function $\boldsymbol{\Delta}_t^{\mathbb{F}} \approx \psi(\boldsymbol{m}_t)$, where the memory vector $\boldsymbol{m}_t$ is defined as $\boldsymbol{m}_t = g_t(\boldsymbol{b}_{t-1}, \boldsymbol{b}_{t-2}, ..., \boldsymbol{b}_1)$. The function $g_t(\cdot)$, which encodes the entire history into the memory vector $\boldsymbol{m}_t$, is computationally intensive. We therefore approximate it using a nested structure, which can be implemented in an iterative form by the memory update function $\phi(\cdot)$:

$$\boldsymbol{m}_t \approx \underbrace{\phi(\phi(\phi(\cdots), \boldsymbol{b}_{t-2}), \boldsymbol{b}_{t-1})}_{t \text{ times}}$$

$$= \phi(\boldsymbol{m}_{t-1}, \boldsymbol{b}_{t-1}). \tag{5}$$

Furthermore, the linear measurement matrix $\mathbf{H}$ defined in Eq. 1, often fails to represent the true observation process. To address this discrepancy, a similar transformation can be made, i.e.,

$$\widetilde{\boldsymbol{b}}_t = \mathbf{H}\boldsymbol{b}_t + \boldsymbol{\Delta}_t^{\mathbb{H}} + \boldsymbol{v}_t, \tag{6}$$

where the mismatch term $\boldsymbol{\Delta}_t^{\mathbb{H}}$ is generated by $\widetilde{\boldsymbol{b}}_t$ through function $\varphi_t(\cdot)$, namely, $\boldsymbol{\Delta}_t^{\mathbb{H}} \approx \varphi_t(\widetilde{\boldsymbol{b}}_t)$.

The memory update function $\phi(\cdot)$, state compensation function $\psi(\cdot)$, and measurement compensation function $\varphi(\cdot)$ are difficult to model with explicit analytical forms. Such that we employ NN technique to fit these complex, non-linear functions. By integrating these learned modules with the foundational principles of Eqs. 5 - 6, we construct a data-driven Bayesian filter: the MeKF, as shown in Figure 3.

### 3.3.1 STRUCTURE OF MEKF

**Memory Update Gate.** The memory update process in Eq. 5 is formally analogous to the Recurrent Neural Network. We therefore implement the update function $\phi(\cdot)$ using the Long Short-Term Memory (LSTM) network. The LSTM is trained to distill and update the memory from the historical trajectory sequence, with the specific update process detailed as follows:

$$\boldsymbol{m}_t = \mathcal{F}_{\text{LSTM}}(\boldsymbol{c}_{t-1}, \boldsymbol{h}_{t-1}, \boldsymbol{m}_{t-1}), \tag{7}$$

where $\mathcal{F}_{\text{LSTM}}(\cdot)$ denotes the mapping function of the MUG, implemented by the LSTM network. And $\boldsymbol{c}_{t-1}$ and $\boldsymbol{h}_{t-1}$ are the cell state and hidden state of the LSTM, respectively.

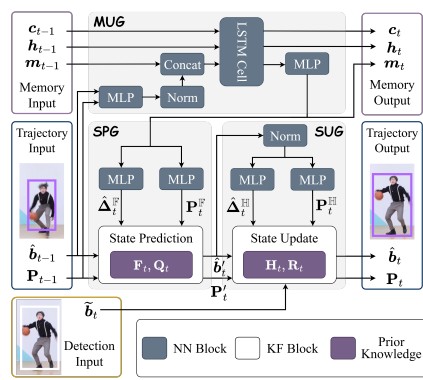

Figure 3: Framework of MeKF.

**State Prediction Gate.** In contrast to MoveSORT and DiffMOT, which directly utilize NN to predict the target's state, the SPG compensates for the error between the physical motion model and the true physical process. While reducing the amount of parameters, the SPG leverages a prior model to guarantee the error lower bound of the MeKF, which is defined as follows:

$$\hat{\boldsymbol{b}}_t' = \mathbf{F}\hat{\boldsymbol{b}}_{t-1} + \hat{\boldsymbol{\Delta}}_t^{\mathbb{F}}, \tag{8}$$

$$\mathbf{P}_t' = \mathbf{F}\mathbf{P}_{t-1}\mathbf{F}^\top + \mathbf{P}_t^{\mathbb{F}} + \mathbf{Q}_t, \tag{9}$$

where $\hat{\boldsymbol{b}}_t'$ and $\mathbf{P}_t'$ represent the state prediction and the error covariance prediction, respectively. Here, $\hat{\boldsymbol{\Delta}}_t^{\mathbb{F}} = \mathcal{F}_{\text{MLP}}^1(\boldsymbol{m}_t)$ and $\mathbf{P}_t^{\mathbb{F}} = \mathcal{F}_{\text{MLP}}^2(\boldsymbol{m}_t)(\mathcal{F}_{\text{MLP}}^2(\boldsymbol{m}_t))^\top$ are the exception and covariance compensation generated by distinct multilayer perceptrons (MLP) with unshared parameters.

**State Update Gate.** Similarly, the SUG utilizes distinct MLPs to generate corresponding compensation terms and is naturally embedded within the state update process, namely,

$$\mathbf{K}_t = \mathbf{P}_t'\mathbf{H}^\top(\mathbf{H}\mathbf{P}_t'\mathbf{H}^\top + \mathbf{P}_t^{\mathbb{H}} + \mathbf{R}_t)^{-1}, \tag{10}$$

$$\hat{\boldsymbol{b}}_t = \hat{\boldsymbol{b}}_t' + \mathbf{K}_t(\widetilde{\boldsymbol{b}}_t - \mathbf{H}\hat{\boldsymbol{b}}_t' - \hat{\boldsymbol{\Delta}}_t^{\mathbb{H}}), \tag{11}$$

$$\mathbf{P}_t = (\mathbf{I} - \mathbf{K}_t\mathbf{H})\mathbf{P}_t', \tag{12}$$

where $\hat{\boldsymbol{b}}_t$ and $\mathbf{P}_t$ are the state update and the error covariance update, respectively. Here, $\hat{\boldsymbol{\Delta}}_t^{\mathbb{H}} = \mathcal{F}_{\text{MLP}}^3(\hat{\boldsymbol{b}}_t')$, $\mathbf{P}_t^{\mathbb{H}} = \mathcal{F}_{\text{MLP}}^4(\hat{\boldsymbol{b}}_t')(\mathcal{F}_{\text{MLP}}^4(\hat{\boldsymbol{b}}_t'))^\top$ and $\mathbf{K}_t$ is the Kalman gain. The derivation is detailed in Appendix B. All of the aforementioned gates are designed based on Bayesian principles similar to the KF and are derived according to Wang et al. (2012).

### 3.3.2 LOSS FUNCTION AND TRAINING PREPARATION

The analytical expression of the MeKF (Eqs. 8 - 12), derived through a Gaussian approximation, renders the filter fully differentiable (Yan et al., 2024), enabling end-to-end training via a loss function composed of mean square error (MSE) and L2 regularization, as calculated below:

$$\mathcal{L} = \frac{1}{JT}\sum_{j=1}^{J}\sum_{t=1}^{T}||\hat{\boldsymbol{b}}_t^j(\widetilde{\boldsymbol{b}}_t^j; \boldsymbol{\Theta}) - \bar{\boldsymbol{b}}_t^j||^2 + \gamma||\boldsymbol{\Theta}||^2, \tag{13}$$

where $\boldsymbol{\Theta}$ represents the set of learnable parameters in the MeKF, and $\gamma$ is the L2 regularization coefficient. The loss is computed over $J$ training sequences in a batch, each of length $T$.

End-to-end training of the MeKF requires a dataset of paired trajectory sequences, each consisting of a detection $\widetilde{\boldsymbol{b}}_t^j$, and its corresponding ground truth box $\bar{\boldsymbol{b}}_t^j$. We construct this dataset by selecting detections from a candidate pool and pairing them with ground truth boxes based on their IoU. A detailed description of this dataset generation procedure is provided in Appendix C.

### 3.4 MOTION-ADAPTIVE ASSOCIATION

To achieve robust association in severe occlusion scenarios, we introduce the Motion-adaptive IoU (Mo-IoU). It is defined as a multiplicative fusion of two IoU variants with an adaptive parameter setting:

$$\text{Mo-IoU}(\hat{\boldsymbol{b}}_t', \widetilde{\boldsymbol{b}}_t, p_t, q_t) = \text{EIoU}(\hat{\boldsymbol{b}}_t', \widetilde{\boldsymbol{b}}_t, p_t) \times \text{HIoU}(\hat{\boldsymbol{b}}_t', \widetilde{\boldsymbol{b}}_t, q_t), \tag{14}$$

where Expansion IoU (EIoU) expands matching region to enhances the probability of establishing reliable matches, and Height IoU (HIoU) emphasizes height similarity to distinguish occluded targets. The parameters $p_t$ and $q_t$ are adaptively set by our Motion-Adaptive Technique (MAT).

**Expansion IoU.** Motivated by C-BIoU (Fan et al., 2023), we design EIoU to relax box boundaries, effectively enlarging the matching region to enhance association likelihood, ultimately leading more continuous target tracking. Formally, EIoU is defined as:

$$\text{EIoU}(\hat{\boldsymbol{b}}_t', \widetilde{\boldsymbol{b}}_t, p_t) = \text{IoU}(\hat{\boldsymbol{e}}_t', \widetilde{\boldsymbol{e}}_t), \tag{15}$$

where $\hat{\boldsymbol{e}}_t' = [\hat{x}_t', \hat{y}_t', (2p_t+1)\hat{w}_t', (2p_t+1)\hat{h}_t']^\top$ and $\widetilde{\boldsymbol{e}}_t = [\widetilde{x}_t, \widetilde{y}_t, (2p_t+1)\widetilde{w}_t, (2p_t+1)\widetilde{h}_t]^\top$ are the expansion boxes of $\hat{\boldsymbol{b}}_t'$ and $\widetilde{\boldsymbol{b}}_t$, respectively. The expansion scaling factor $p_t$ controls the expansion scale of the boxes. When $p_t=0$, no expansion occurs, and EIoU degenerates to the standard IoU.

**Height IoU.** Recognizing that height remains a highly distinguishable feature under severe occlusion, we introduce HIoU, inspired by Hybrid-SORT (Yang et al., 2024), to reinforce height similarity and mitigate the ambiguity potentially induced by EIoU. And HIoU is defined as:

$$\text{HIoU}(\hat{\boldsymbol{b}}_t', \widetilde{\boldsymbol{b}}_t, q_t) = \left( \frac{l_t}{\hat{h}_t' + \widetilde{h}_t - l_t} \right)^{q_t}, \tag{16}$$

where $l_t$ denotes the intersection height of $\hat{\boldsymbol{b}}_t'$ and $\widetilde{\boldsymbol{b}}_t$, and the exponent $q_t$ adaptively controls the emphasis placed on this height similarity. The base of this formula is geometrically equivalent to a 1D-IoU on the vertical axis, robustly measuring the boxes' vertical alignment.

**Motion-Adaptive Technique.** To improve the generalization of Mo-IoU in diverse scenarios, a novel MAT is proposed to adaptively adjust the expansion scaling parameter $p_t$ and the height modulation parameter $q_t$ based on the target's motion characteristics, as formulated below:

$$p_t = \begin{cases} M_{\text{slow}} & \text{if } \dot{c}_{t-1} \leq \Theta_{\text{center}}, \\ M_{\text{fast}} & \text{otherwise}. \end{cases} \tag{17} \qquad q_t = \begin{cases} N_{\text{slow}} & \text{if } \dot{l}_{t-1} \leq \Theta_{\text{height}}, \\ N_{\text{fast}} & \text{otherwise}. \end{cases} \tag{18}$$

where $\dot{c}_{t-1} = \sqrt{(\dot{x}_{t-1}/w_{t-1})^2 + (\dot{y}_{t-1}/h_{t-1})^2}$ and $\dot{l}_{t-1} = \dot{h}_{t-1}/h_{t-1}$ represent the normalized speeds of the box center and height, respectively, with a dot denoting velocity. The terms $\Theta_{\text{center}}$ and $\Theta_{\text{height}}$ are predefined thresholds for these two speeds. Instead of continuously tuning $p_t$ and $q_t$, which would be computationally expensive, we adopt a discrete piecewise design. This choice strikes a balance between adaptivity and efficiency, ensuring practical applicability in real-time tracking. As a scale-invariant metric, the normalized speed is a suitable quantitative description of the target's motion characteristics.

The parameter $p_t$ compensates for the motion model's prediction error. Since high-speed motion often leads to larger errors, a larger expansion scaling parameter ($p_t=M_{\text{fast}}$) is used to provide greater spatial tolerance, and vice versa. In contrast, the parameter $q_t$ adapts to the reliability of height as a feature: a rapidly changing, less reliable height warrants a smaller height modulation parameter ($q_t=N_{\text{fast}}$), and vice versa.

## 4 EXPERIMENTS

### 4.1 DATASETS AND METRICS

**Datasets.** We conducted the main experiments on DanceTrack and SportsMOT datasets known for their diverse and rapid movements and indistinguishable appearances, in which the performance of ReID module is highly limited, requiring accurate motion capability. DanceTrack features severe occlusion and similar appearance, demanding robust motion capacity for long-term identity consistency. SportsMOT introduces fast, variable-speed target motion and extensive camera motion, requiring more robust motion models and association.

Furthermore, we conducted comparative experiments on the MOT17 and MOT20 datasets, which are characterized by relatively linear and stable motion (Hu et al., 2024). The detailed results have been added to Appendix D.1.

**Metrics.** We utilize Higher Order Metric (Luiten et al., 2021) (HOTA, AssA, DetA), IDF1 (Ristani et al., 2016), and CLEAR metrics (Bernardin & Stiefelhagen, 2008) (MOTA) as our evaluation metrics. Among various metrics, HOTA is the core benchmark that holistically balances association consistency and positional precision. Complementing this, IDF1 and AssA specifically measure association quality and identity preservation, while DetA and MOTA primarily evaluate state estimation accuracy. Additionally, computational efficiency is quantified through frames per second (FPS) to evaluate real-time processing capability.

### 4.2 IMPLEMENTATION DETAILS

For the training of our proposed MeKF, we utilize AdamW optimizer with learning rate set to $10^{-4}$, and regularization coefficient $\gamma$ is set to $0.02$. The hidden size of LSTM cell and MLPs is set to 32, and the state transition matrix $\mathbf{F}$ is set to a constant velocity model. For Mo-IoU, the expansion scaling parameters are set to $M_{\text{slow}}=0.5$ and $M_{\text{fast}}=M_{\text{slow}}+0.1$, while the height modulation parameter are set to $N_{\text{slow}}=2$, with $N_{\text{fast}}=N_{\text{slow}}-1$. Velocity thresholds $\Theta_{\text{center}}$ and $\Theta_{\text{height}}$ are determined by the 70th and 50th percentile of the normalized velocity distribution from training set (i.e. 0.0406 and 0.0090 for DanceTrack, 0.1172 and 0.0062 for SportsMOT).

For the detector, we fine-tune the COCO-pretrained YOLOX-X model on CrowdHuman (Shao et al., 2018) and the target dataset, same to the training procedure used in SportsMOT. In the association stage, the confidence threshold of high-score and low-score matching are set to 0.6 and 0.1. For ReID model, we utilize SBS50 from the fast-reid library (He et al., 2020).

Experiments are conducted on 8 GeForce RTX 4090, while FPS is evaluated in FP16 precision with batchsize of 1 using a single RTX 4090.

### 4.3 BENCHMARK RESULTS

**DanceTrack.** As depicted in Table 1, MeMoSORT establishes a new SOTA on the challenging DanceTrack test set with 67.9% HOTA score. MeMoSORT significantly outperforms traditional KF-based trackers, demonstrating the advantages of the proposed MeKF. In contrast to sliding window-based filters like DiffMOT, which estimate the current state from a fixed-length trajectory history, our method shows superior tracking performance. When compared to other implicit memory-based filters such as TrackSSM, MeMoSORT's hybrid design of physical prior and NN proves more effective than purely data-driven alternatives. By retaining the robust inductive bias of a classic Bayesian filter while using the memory network to handle non-Markovian dynamics, our method achieves a more stable and accurate state estimation. Furthermore, when compared with trackers utilizing similar modified IoU metrics, such as Hybrid-SORT, our synergistic combination of an advanced filter and an adaptive association metric secures a clear performance advantage.

Finally, even against transformer-based end-to-end methods like MeMOTR, our method demonstrates significant advantages in both estimation accuracy and inference speed. Although there is a slight performance gap compared to MOTRv2 and MOTIP, our method inherits the low computational overhead characteristic of the TBD paradigm, offering a substantial advantage in inference speed over these computationally heavy transformer-based trackers.

Table 1: Performance on DanceTrack test set, with FPS evaluated on the validation set. The best/second results are shown in **bold**/underlined.

| Methods | IoU modified | HOTA↑ | AssA↑ | IDF1↑ | DetA↑ | MOTA↑ | FPS↑ |
|---|---|---|---|---|---|---|---|
| *End-to-end tracker:* | | | | | | | |
| MOTRv2 (Zhang et al., 2023) | | 69.9 | 59.0 | 71.7 | 83.0 | 91.9 | 10.2 |
| MeMOTR (Gao & Wang, 2023) | | 63.4 | 52.3 | 65.5 | 77.0 | 85.4 | 17.7 |
| MOTIP (Gao et al., 2025) | | 69.6 | 60.4 | 74.7 | 80.4 | 90.6 | 19.8 |
| *KF-based filter:* | | | | | | | |
| ByteTrack (Zhang et al., 2022) | | 47.7 | 32.1 | 53.9 | 71.0 | 89.6 | **35.8** |
| OC-SORT (Maggiolino et al., 2023) | | 55.1 | 40.4 | 54.9 | 80.4 | 92.2 | - |
| Deep OC-SORT (Maggiolino et al., 2023) | | 61.3 | 45.8 | 61.5 | 82.2 | 92.3 | - |
| TrackTrack (Shim et al., 2025) | | 66.5 | 52.9 | 67.8 | - | **93.6** | - |
| C-BIoU (Fan et al., 2023) | ✓ | 60.6 | 45.4 | 61.6 | 81.3 | 91.6 | - |
| Hybrid-SORT (Yang et al., 2024) | ✓ | 65.7 | - | 67.4 | - | 91.8 | 15.5 |
| *Sliding window-based filter:* | | | | | | | |
| MotionTrack (Xiao et al., 2024b) | | 58.2 | 41.7 | 58.6 | 81.4 | 91.3 | - |
| DiffMOT (Lv et al., 2024) | | 62.3 | 47.2 | 63.0 | 82.5 | 92.8 | 22.7 |
| *Implicit memory-based filter:* | | | | | | | |
| MambaMOT (Huang et al., 2024a) | | 56.1 | 39.0 | 54.9 | 80.8 | 90.3 | 28.8 |
| Track SSM (Hu et al., 2024) | | 57.7 | 41.0 | 57.5 | 81.5 | 92.2 | 20.3 |
| DeepMove SORT (Adžemović et al., 2024) | ✓ | 63.0 | 48.6 | 65.0 | 82.0 | 92.6 | - |
| **MeMoSORT** (ours) | ✓ | **67.9** | **54.3** | **68.0** | **85.0** | 93.4 | 28.8 |

**SportsMOT.** On the SportsMOT benchmark, characterized by fast and variable motion, MeMo-SORT again establishes a new SOTA, as shown in Table 2. This result underscores the superiority of memory-based filters over traditional KF and sliding-window approaches for handling complex dynamics. Within the implicit memory-based paradigm, MeMoSORT's hybrid design further distinguishes it; instead of fully replacing the motion model, our MeKF uses memory to explicitly correct a physics-based prior, leading to more stable and accurate state estimation. Furthermore, by adaptively adjust its parameters, our Mo-IoU robustly resolves ambiguities during severe occlusions, a key factor in its superior performance over other modified IoU techniques.

Table 2: Performance comparison on the SportsMOT test set. The best/second results are shown in **bold**/underlined.

| Methods | IoU modified | HOTA↑ | AssA↑ | IDF1↑ | DetA↑ | MOTA↑ |
|---|---|---|---|---|---|---|
| *Without filter:* | | | | | | |
| Deep-EIoU (Maggiolino et al., 2023) | ✓ | 77.2 | 67.7 | 79.8 | 88.2 | 96.3 |
| Deep HM-SORT (Gran-Henriksen et al., 2024) | ✓ | 80.1 | 72.7 | 85.2 | 88.3 | 96.6 |
| *KF-based filter:* | | | | | | |
| ByteTrack (Zhang et al., 2022) | | 64.1 | 52.3 | 71.4 | 78.5 | 95.9 |
| OC-SORT (Cao et al., 2023) | | 73.7 | 61.5 | 74.0 | 88.5 | 96.5 |
| *Sliding window-based filter:* | | | | | | |
| MotionTrack (Xiao et al., 2024b) | | 74.0 | 61.7 | 74.0 | 88.8 | 96.6 |
| DiffMOT (Lv et al., 2024) | | 76.2 | 65.1 | 76.1 | 89.3 | **97.1** |
| *Implicit memory-based filter:* | | | | | | |
| MambaMOT (Huang et al., 2024a) | | 71.3 | 58.6 | 71.1 | 86.7 | 94.9 |
| Track SSM (Hu et al., 2024) | | 74.4 | 62.4 | 74.5 | 88.8 | 96.8 |
| SportMamba (Khanna et al., 2025) | ✓ | 77.3 | 66.8 | 77.7 | **89.5** | 96.9 |
| DeepMove SORT (Adžemović et al., 2024) | ✓ | 78.7 | 70.3 | 81.7 | 88.1 | 96.5 |
| MeMoSORT(ours) | ✓ | **82.1** | **75.6** | **86.4** | 89.3 | 97.0 |

## 4.4 ABLATION STUDY

We conduct ablation studies on the DanceTrack validation set, which concentrate on investigating the impact of different components, different filters, different training data quality, and different IoU variants on the proposed MeMoSORT.

**Component Ablation.** The proposed MeMoSORT algorithm comprises two components, MeKF and Mo-IoU, whose individual contributions are examined through ablation studies, as the results

shown in Table 3. Using ByteTrack as the baseline (line 1), we first replace its KF with MeKF (line 2), which yields a significant gain and confirms that the non-Markovian modeling improves motion prediction and filtering. Next, we substitute the baseline association module with Mo-IoU (line 3), improving association and thus HOTA. When both modules are combined (line 4), performance is further boosted by jointly enhancing state estimation and association. Finally, adding ReID information alongside Mo-IoU (line 5) brings additional slight gains, though with a drop in FPS. We attribute this modest gain to the degradation of the ReID model in challenging scenes with severe occlusions, which causes target appearance to become indistinguishable.

Table 3: Ablation study of MeMoSORT's key components on the DanceTrack validation set. The best/second results are shown in **bold**/underlined.

| MeKF | Mo-IoU | ReID | HOTA ↑ | AssA ↑ | IDF1 ↑ | DetA ↑ | MOTA ↑ | FPS ↑ |
|------|--------|------|--------|--------|--------|--------|--------|-------|
|      |        |      | 56.94 | 34.92 | 48.18 | 92.91 | 96.35 | **74.5** |
| ✓    |        |      | 67.41 | 49.58 | 66.41 | 91.69 | 97.55 | 60.8 |
|      | ✓      |      | 68.32 | 50.35 | 63.86 | 92.76 | 97.30 | 62.0 |
| ✓    | ✓      |      | 77.54 | 64.73 | 76.92 | 92.93 | **97.74** | 49.4 |
| ✓    | ✓      | ✓    | **77.91** | **65.21** | **77.49** | **93.13** | 97.73 | 28.8 |

**Performance with Different Filter.** Noting that the proposed tracking framework leverages MeKF to enhance motion prediction and update, thereby improving overall tracking performance, we further compare MeKF against other filtering methods within the same ByteTrack baseline, as the results shown in Table 4. Specifically, we replace the baseline filter with several standard KFs using different motion models, as well as several data-driven filters, while keeping ByteTrack's association module unchanged. Results show that MeKF consistently achieves the best performance across most metrics, demonstrating superior state estimation accuracy through its non-Markovian modeling. The NN blocks in MeKF assist the physical motion model by generating compensation for its errors, based on memory and detection respectively.

Compared with the KF that uses a constant velocity (CV) motion model, using more complex motion models, such as constant acceleration (CA), coordinated turn (CT), and constant turn rate and acceleration (CTRA), does not lead to significant performance gains. This is because all these models are based on the first-order Markovian assumption, making them fail to adapt to the complex, non-Markovian motion patterns inherent in the DanceTrack dataset. Furthermore, compared with data-driven methods, the MeKF robustly ensures the stability of the state estimation; even if the NN fails, the underlying physical model can still provide a baseline prediction as a failsafe.

Table 4: Performance comparison of different filter on the DanceTrack validation set. The best/second results are shown in **bold**/underlined.

| Filter | HOTA ↑ | AssA ↑ | IDF1 ↑ | DetA ↑ | MOTA ↑ |
|--------|--------|--------|--------|--------|--------|
| KF (CV) | 56.94 | 34.92 | 48.18 | 92.91 | 96.35 |
| KF (CA) | 57.55 | 39.46 | 54.64 | 84.18 | 92.98 |
| KF (CT) | 57.32 | 39.21 | 54.25 | 84.05 | 93.08 |
| KF (CTRA) | 58.13 | 40.43 | 55.54 | 83.83 | 93.09 |
| LSTM (Hochreiter & Schmidhuber, 1997) | 60.16 | 38.97 | 52.31 | 92.94 | 96.64 |
| Transformer (Vaswani et al., 2017) | 64.12 | 44.20 | 57.60 | **93.08** | 97.04 |
| Diffusion (Lv et al., 2024) | 65.91 | 46.78 | 60.38 | 92.93 | 97.15 |
| **MeKF** (ours) | **67.41** | **49.58** | **66.41** | 91.69 | **97.55** |

**Performance with Different Training Data Quality.** As the training process of MeKF relies on detection data to learn and generate compensations for observation process mismatches, we conducted an ablation study on DanceTrack to investigate the sensitivity of MeKF to detector quality. Specifically, following the consistent training pipeline described in Section 4.2, we trained YOLOX models of varying scales (YOLOX-S, -M, and -L) to generate distinct training datasets of differing quality and retrained the MeKF accordingly. The comparison results of MeKFs within the same ByteTrack baseline are presented in Table 5.

Notably, MeKF retains strong performance even with the lightweight YOLOX-S. Despite the significant drop in detection precision, the MeKF effectively learns to compensate for the higher ob-

servation bias. This demonstrates that our method is not strictly dependent on high-quality inputs; rather, it adaptively models the specific performance of the detector.

Table 5: Sensitivity analysis on detector dependency for MeKF training. The best/second results are shown in **bold**/underlined.

| Detector | HOTA ↑ | AssA ↑ | IDF1 ↑ | DetA ↑ | MOTA ↑ |
|----------|--------|--------|--------|--------|--------|
| YOLOX-S | 59.51 | 42.51 | 60.71 | 83.49 | 93.63 |
| YOLOX-M | 63.22 | 44.85 | 62.34 | 89.20 | 96.31 |
| YOLOX-L | 65.19 | 47.26 | 64.06 | 89.98 | 96.88 |
| YOLOX-X | **67.41** | **49.58** | **66.41** | **91.69** | **97.55** |

**Performance with Different IoU Variants.** In Table 6, we compare the performance of different association methods, where the motion prediction and update components are consistently handled by MeKF. HMIoU, proposed in Hybrid-SORT, combines IoU with HIoU to incorporate height similarity, while HA-EIoU, introduced in SportMamba, multiplies EIoU with HIoU to enhance association performance. Our proposed Mo-IoU achieves the best results across all metrics, outperforming existing IoU variants. Its superior performance can be attributed to its adaptive parameter selection, which jointly controls the expansion scale and height weighting, resulting in more robust and accurate tracking. Moreover, the HIoU introduced in Mo-IoU counterbalances the looseness of EIoU, yielding a significant improvement in association robustness compared to EIoU alone.

Table 6: Performance comparison of different IoU variants on the DanceTrack validation set.

| IoU variants | HOTA ↑ | AssA ↑ | IDF1 ↑ | DetA ↑ | MOTA ↑ |
|--------------|--------|--------|--------|--------|--------|
| IoU (Yu et al., 2016) | 67.41 | 49.58 | 66.41 | 91.69 | 97.55 |
| EIoU (Fan et al., 2023) | 70.80 | 54.37 | 70.50 | 92.24 | 97.62 |
| HMIoU (Yang et al., 2024) | 72.70 | 57.15 | 71.65 | 92.52 | 97.66 |
| HA-EIoU (Khanna et al., 2025) | 75.21 | 60.97 | 74.53 | 92.81 | 97.71 |
| Mo-IoU(ours) | **77.54** | **64.73** | **76.92** | **92.93** | **97.74** |

## 5 CONCLUSION

In this paper, we present MeMoSORT, a simple, online and real-time MOT algorithm designed to overcome key limitations in conventional TBD methods. Our approach introduces two key innovations: the MeKF, which uses a memory-augmented NN to correct state estimation errors, and the Mo-IoU, which adaptively expands the matching region and incorporates height similarity to ensure robust association. The effectiveness of our method is demonstrated through extensive experiments, where MeMoSORT achieves SOTA performance on the challenging benchmark DanceTrack and SportsMOT, providing a robust solution for MOT challenges.

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

APPENDIX

## Table of Contents

## A   ANALYSIS OF NON-MARKOVIAN DYNAMICS IN TARGET TRAJECTORIES

Conventional KF-based MOT algorithms typically adopt a first-order Markov assumption to simplify target dynamics. However, real-world targets often exhibit more complex motion with long-term temporal correlations, as illustrated in Figure 4, a phenomenon we refer to as non-Markovian dynamics.

As shown in Figure 4(a), a visual inspection of the target's trajectory strongly suggests its motion has significant non-Markovian properties. The path is not a simple random walk but can be decomposed into three distinct phases: an initial period of localized, high-frequency movement (yellow area); a middle phase of directional, long-range displacement (pink area); and a final phase of dense hovering in a new local area (purple area). This phased switching from a stable local pattern to a directional journey and back again strongly implies an underlying "plan" or "intent" that a memoryless Markovian model could not produce. Furthermore, the high degree of path overlap and repeated visits to specific areas demonstrate a form of memory, directly contradicting the core Markovian assumption that the future depends only on the present. In summary, the trajectory's clear structure, apparent purposefulness, and historical dependence provide strong qualitative evidence of its non-Markovian nature.

The trajectory shown in Figure 4(b) provides even more compelling evidence of non-Markovian dynamics. It moves in a predictable, back-and-forth pattern, creating a clear rhythm. This is the

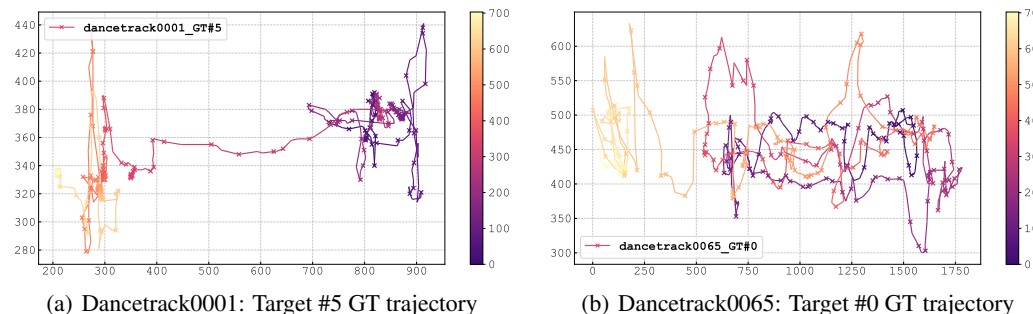

(a) Dancetrack0001: Target #5 GT trajectory  (b) Dancetrack0065: Target #0 GT trajectory

Figure 4: Two representative ground truth (GT) trajectories from the DanceTrack dataset, showcasing complex and non-Markovian motion. The color of the path indicates the progression of time, evolving from purple (start) to yellow (end).The x-axis and y-axis represent the target positions in image coordinates (pixels).

opposite of a chaotic random walk. This pattern is not static; it displays multi-scale dynamics, with the amplitude and frequency of the oscillations evolving throughout the sequence. Such a structured and evolving "choreography" points to a process with significant state memory.

The non-Markovian nature is further confirmed by the trajectory's continuity across interruptions. When the target reappears after a gap in observation, its motion pattern seamlessly resumes rather than resetting to a random state. This suggests a persistent "intent" that violates the core memoryless assumption of the Markov process.

## B    DERIVATION OF MEKF

### B.1    BAYESIAN FILTERS FOR NON-MARKOVIAN PROCESSES

Before deriving the analytical expression for our MeKF, we first establish a general Bayesian filtering framework for non-Markovian dynamics to describe the computation of the relevant probability density functions (PDFs). Within this framework, obtaining the filtered estimate at time step $t$ requires computing the joint posterior PDF of the entire history of target states $\boldsymbol{b}_{1:t}$ and memory $\boldsymbol{m}_{1:t}$. This is conditioned on all available measurements up to the current time, namely, $\widetilde{\boldsymbol{b}}_{1:t}$, as well as the training data $\mathcal{D}$ (the detailed generation procedure for this dataset is described in Appendix C). Formally, the density of interest is $p(\boldsymbol{b}_{1:t}, \boldsymbol{m}_{1:t}|\widetilde{\boldsymbol{b}}_{1:t}, \mathcal{D})$.

According to Bayes' theorem, this posterior probability density can be decomposed as follows:

$$
\begin{aligned}
p(\boldsymbol{b}_{1:t}, \boldsymbol{m}_{1:t}|\widetilde{\boldsymbol{b}}_{1:t}, \mathcal{D}) &= p(\boldsymbol{b}_{1:t}, \boldsymbol{m}_{1:t}|\widetilde{\boldsymbol{b}}_{1:t-1}, \widetilde{\boldsymbol{b}}_t, \mathcal{D}) \\
&= \frac{p(\widetilde{\boldsymbol{b}}_t|\boldsymbol{b}_{1:t}, \boldsymbol{m}_{1:t}, \widetilde{\boldsymbol{b}}_{1:t-1}, \mathcal{D})p(\boldsymbol{b}_{1:t}, \boldsymbol{m}_{1:t}|\widetilde{\boldsymbol{b}}_{1:t-1}, \mathcal{D})}{p(\widetilde{\boldsymbol{b}}_t|\widetilde{\boldsymbol{b}}_{1:t-1}, \mathcal{D})} \quad (19) \\
&\propto p(\widetilde{\boldsymbol{b}}_t|\boldsymbol{b}_{1:t}, \boldsymbol{m}_{1:t}, \widetilde{\boldsymbol{b}}_{1:t-1}, \mathcal{D})p(\boldsymbol{b}_{1:t}, \boldsymbol{m}_{1:t}|\widetilde{\boldsymbol{b}}_{1:t-1}, \mathcal{D}). \quad (20)
\end{aligned}
$$

Since the detection $\widetilde{\boldsymbol{b}}_t$ is generated by the detector based only on the current ground truth state $\boldsymbol{b}_t$, it is independent of the memory $\boldsymbol{m}_{1:t}$. Consequently, the corresponding likelihood PDF can be expressed equivalently as:

$$
p(\widetilde{\boldsymbol{b}}_t|\boldsymbol{b}_{1:t}, \boldsymbol{m}_{1:t}, \widetilde{\boldsymbol{b}}_{1:t-1}, \mathcal{D}) = p(\widetilde{\boldsymbol{b}}_t|\boldsymbol{b}_t, \mathcal{D}). \quad (21)
$$

To account for the observation model mismatch present in Eq. 6, we express the likelihood PDF in the following integral form:

$$
\begin{aligned}
p(\widetilde{\boldsymbol{b}}_t|\boldsymbol{b}_t, \mathcal{D}) &= \int p(\widetilde{\boldsymbol{b}}_t, \boldsymbol{\Delta}_t^{\mathbb{H}}|\boldsymbol{b}_t, \mathcal{D})\mathrm{d}\boldsymbol{\Delta}_t^{\mathbb{H}} \\
&= \int p(\widetilde{\boldsymbol{b}}_t|\boldsymbol{\Delta}_t^{\mathbb{H}}, \boldsymbol{b}_t, \mathcal{D})p(\boldsymbol{\Delta}_t^{\mathbb{H}}|\boldsymbol{b}_t, \mathcal{D})\mathrm{d}\boldsymbol{\Delta}_t^{\mathbb{H}}. \quad (22)
\end{aligned}
$$

According to the total probability formula, the prior PDF in Eq. 19 can be expressed as follows:

$$p(\boldsymbol{b}_{1:t}, \boldsymbol{m}_{1:t}|\widetilde{\boldsymbol{b}}_{1:t-1}, \mathcal{D}) = p(\boldsymbol{b}_t, \boldsymbol{m}_t|\boldsymbol{b}_{1:t-1}, \boldsymbol{m}_{1:t-1}, \widetilde{\boldsymbol{b}}_{1:t-1}, \mathcal{D})p(\boldsymbol{b}_{1:t-1}, \boldsymbol{m}_{1:t-1}|\widetilde{\boldsymbol{b}}_{1:t-1}, \mathcal{D}). \quad (23)$$

The second term on the right-hand side of Eq. 23 is the joint posterior PDF of the state and memory at time $t-1$, while the term on the left-hand side represents the joint transition process for the state and memory that captures the system's non-Markovian dynamics. Applying the conditional independence expressed by Eqs. 4 and 5, this transition process can be expressed as follows:

$$\begin{aligned}
&p(\boldsymbol{b}_t, \boldsymbol{m}_t|\boldsymbol{b}_{1:t-1}, \boldsymbol{m}_{1:t-1}, \widetilde{\boldsymbol{b}}_{1:t-1}, \mathcal{D}) \\
&= p(\boldsymbol{b}_t|\boldsymbol{m}_t, \boldsymbol{b}_{1:t-1}, \boldsymbol{m}_{1:t-1}, \widetilde{\boldsymbol{b}}_{1:t-1}, \mathcal{D})p(\boldsymbol{m}_t|\boldsymbol{b}_{1:t-1}, \boldsymbol{m}_{1:t-1}, \widetilde{\boldsymbol{b}}_{1:t-1}, \mathcal{D}) \\
&= \int p(\boldsymbol{b}_t|\boldsymbol{\Delta}_t^{\mathbb{F}}, \boldsymbol{m}_t, \boldsymbol{b}_{1:t-1}, \boldsymbol{m}_{1:t-1}, \widetilde{\boldsymbol{b}}_{1:t-1}, \mathcal{D})p(\boldsymbol{\Delta}_t^{\mathbb{F}}|\boldsymbol{m}_t, \boldsymbol{b}_{1:t-1}, \boldsymbol{m}_{1:t-1}, \widetilde{\boldsymbol{b}}_{1:t-1}, \mathcal{D}) \\
&\quad \times p(\boldsymbol{m}_t|\boldsymbol{b}_{t-1}, \boldsymbol{m}_{t-1}, \mathcal{D})\mathrm{d}\boldsymbol{\Delta}_t^{\mathbb{F}} \\
&= \int p(\boldsymbol{b}_t|\boldsymbol{\Delta}_t^{\mathbb{F}}, \boldsymbol{b}_{t-1}, \mathcal{D})p(\boldsymbol{\Delta}_t^{\mathbb{F}}|\boldsymbol{m}_t, \mathcal{D})p(\boldsymbol{m}_t|\boldsymbol{b}_{t-1}, \boldsymbol{m}_{t-1}, \mathcal{D})\mathrm{d}\boldsymbol{\Delta}_t^{\mathbb{F}}. \quad (24)
\end{aligned}$$

Based on the Bayesian theorem, the joint posterior of state and memory can be obtained as:

$$\begin{aligned}
p(\boldsymbol{b}_{1:t}, \boldsymbol{m}_{1:t}|\widetilde{\boldsymbol{b}}_{1:t}, \mathcal{D}) \propto &\int p(\widetilde{\boldsymbol{b}}_t|\boldsymbol{\Delta}_t^{\mathbb{H}}, \boldsymbol{b}_t, \mathcal{D})p(\boldsymbol{\Delta}_t^{\mathbb{H}}|\boldsymbol{b}_t, \mathcal{D})\mathrm{d}\boldsymbol{\Delta}_t^{\mathbb{H}} \\
&\times \int p(\boldsymbol{b}_t|\boldsymbol{\Delta}_t^{\mathbb{F}}, \boldsymbol{b}_{t-1}, \mathcal{D})p(\boldsymbol{\Delta}_t^{\mathbb{F}}|\boldsymbol{m}_t, \mathcal{D})p(\boldsymbol{m}_t|\boldsymbol{b}_{t-1}, \boldsymbol{m}_{t-1}, \mathcal{D})\mathrm{d}\boldsymbol{\Delta}_t^{\mathbb{F}} \\
&\times p(\boldsymbol{b}_{1:t-1}, \boldsymbol{m}_{1:t-1}|\widetilde{\boldsymbol{b}}_{1:t-1}, \mathcal{D}). \quad (25)
\end{aligned}$$

### B.2 IMPLEMENTATION WITH GAUSSIAN APPROXIMATION

While the above derivation establishes the general Bayesian filtering framework, its direct implementation involves various methods. For the purposes of computational efficiency and stability, we choose to implement the framework using Gaussian approximation. The following assumptions are therefore required to perform this approximation.

**Assumption 1.** The process noise $\boldsymbol{w}_t$ given in Eq. 4 obeys Gaussian distribution with a mean of $\boldsymbol{0}$ and a covariance of $\mathbf{Q}_t$, namely, $\boldsymbol{w}_t \sim \mathcal{N}(\boldsymbol{0}, \mathbf{Q}_t)$. And the measurement noise $\boldsymbol{v}_t$ given in Eq. 6 obeys a Gaussian distribution with a mean of $\boldsymbol{0}$ and a covariance of $\mathbf{R}_t$, namely, $\boldsymbol{v}_t \sim \mathcal{N}(\boldsymbol{0}, \mathbf{R}_t)$.

**Assumption 2.** The state posterior PDF obeys a Gaussian distribution with first- and second-order moments of $\hat{\boldsymbol{b}}_t$ and $\mathbf{P}_t$, respectively, namely,

$$p(\boldsymbol{b}_{1:t}|\widetilde{\boldsymbol{b}}_{1:t}, \mathcal{D}) = \mathcal{N}(\boldsymbol{b}_t; \hat{\boldsymbol{b}}_t, \mathbf{P}_t). \quad (26)$$

**Assumption 3.** The state transition mismatch term $\boldsymbol{\Delta}_t^{\mathbb{F}}$ obeys a Gaussian distribution with first- and second-order moments of $\hat{\boldsymbol{\Delta}}_t^{\mathbb{F}}$ and $\mathbf{P}_t^{\mathbb{F}}$, respectively. And the observation mismatch term $\boldsymbol{\Delta}_t^{\mathbb{H}}$ obeys a Gaussian distribution with first- and second-order moments of $\hat{\boldsymbol{\Delta}}_t^{\mathbb{H}}$ and $\mathbf{P}_t^{\mathbb{H}}$, respectively, namely,

$$p(\boldsymbol{\Delta}_t^{\mathbb{F}}|\boldsymbol{c}_t, \mathcal{D}) = \mathcal{N}(\boldsymbol{\Delta}_t^{\mathbb{F}}; \hat{\boldsymbol{\Delta}}_t^{\mathbb{F}}, \mathbf{P}_t^{\mathbb{F}}), \quad (27)$$

$$p(\boldsymbol{\Delta}_t^{\mathbb{H}}|\boldsymbol{b}_t, \mathcal{D}) = \mathcal{N}(\boldsymbol{\Delta}_t^{\mathbb{H}}; \hat{\boldsymbol{\Delta}}_t^{\mathbb{H}}, \mathbf{P}_t^{\mathbb{H}}). \quad (28)$$

#### B.2.1 IMPLEMENTATION FOR STATE PREDICTION

Based on Eq. 4, the mean of state prediction is calculated as:

$$\begin{aligned}
\hat{\boldsymbol{b}}_t' &= \mathbb{E}_{p(\boldsymbol{b}_{1:t}|\widetilde{\boldsymbol{b}}_{1:t-1}, \mathcal{D})}\{\boldsymbol{b}_t\} \\
&= \mathbb{E}_{p(\boldsymbol{b}_{1:t}|\widetilde{\boldsymbol{b}}_{1:t-1}, \mathcal{D})}\{\mathbf{F}\boldsymbol{b}_{t-1} + \boldsymbol{\Delta}_t^{\mathbb{F}} + \boldsymbol{w}_t\} \\
&= \iiiint (\mathbf{F}\boldsymbol{b}_{t-1} + \boldsymbol{\Delta}_t^{\mathbb{F}}) P_t^1 \mathrm{d}\boldsymbol{\Delta}_t^{\mathbb{F}}\mathrm{d}\boldsymbol{m}_t\mathrm{d}\boldsymbol{m}_{t-1}\mathrm{d}\boldsymbol{b}_{t-1}, \quad (29)
\end{aligned}$$

where $P_t^1 = p(\mathbf{\Delta}_t^{\mathbb{F}}|\boldsymbol{m}_t, \mathcal{D})p(\boldsymbol{m}_t|\boldsymbol{b}_{t-1}, \boldsymbol{m}_{t-1}, \mathcal{D})p(\boldsymbol{b}_{1:t-1}, \boldsymbol{m}_{1:t-1}|\widetilde{\boldsymbol{b}}_{1:t-1}, \mathcal{D})$.

According to Eq. 26, the state posterior PDF at time $t-1$ is formulated as:

$$p(\boldsymbol{b}_{1:t-1}|\widetilde{\boldsymbol{b}}_{1:t-1}, \mathcal{D}) = \mathcal{N}(\boldsymbol{b}_{t-1}; \hat{\boldsymbol{b}}_{t-1}, \mathbf{P}_{t-1}). \tag{30}$$

Substituting Eq. 30 and the Eq. 27 into Eq. 29, the analytical expression of state prediction mean can be calculated as:

$$\hat{\boldsymbol{b}}_t' = \mathbf{F}\hat{\boldsymbol{b}}_{t-1} + \hat{\mathbf{\Delta}}_t^{\mathbb{F}}. \tag{31}$$

The state prediction covariance is calculated as:

$$\begin{aligned}
\mathbf{P}_t' &= \mathbb{E}_{p(\boldsymbol{b}_{1:t}|\widetilde{\boldsymbol{b}}_{1:t-1}, \mathcal{D})}\left\{ \left(\boldsymbol{b}_t - \hat{\boldsymbol{b}}_t'\right)\left(\boldsymbol{b}_t - \hat{\boldsymbol{b}}_t'\right)^\top \right\} \\
&= \iiiint \left(\mathbf{F}\boldsymbol{b}_{t-1} + \mathbf{\Delta}_t^{\mathbb{F}} + \boldsymbol{w}_t - \hat{\boldsymbol{b}}_t'\right)\left(\mathbf{F}\boldsymbol{b}_{t-1} + \mathbf{\Delta}_t^{\mathbb{F}} + \boldsymbol{w}_t - \hat{\boldsymbol{b}}_t'\right)^\top P_t^1 \mathrm{d}\mathbf{\Delta}_t^{\mathbb{F}}\mathrm{d}\boldsymbol{m}_t\mathrm{d}\boldsymbol{m}_{t-1}\mathrm{d}\boldsymbol{b}_{t-1}.
\end{aligned} \tag{32}$$

Substituting Eq. 27 and Eq. 30 into Eq. 32, thus we have the state prediction covariance as follows:

$$\mathbf{P}_t' = \mathbf{F}\mathbf{P}_{t-1}\mathbf{F}^\top + \mathbf{P}_t^{\mathbb{F}} + \mathbf{Q}_t. \tag{33}$$

### B.2.2 IMPLEMENTATION FOR STATE UPDATE

According to Eqs. 6 and 28, the mean value of the measurement prediction is calculated as:

$$\begin{aligned}
\widetilde{\boldsymbol{b}}_t' &= \mathbb{E}_{p(\widetilde{\boldsymbol{b}}_t|\widetilde{\boldsymbol{b}}_{1:t-1}, \mathcal{D})}\left\{ \widetilde{\boldsymbol{b}}_t \right\} \\
&= \mathbb{E}_{p(\widetilde{\boldsymbol{b}}_t|\widetilde{\boldsymbol{b}}_{1:t-1}, \mathcal{D})}\left\{ \mathbf{H}\boldsymbol{b}_t + \mathbf{\Delta}_t^{\mathbb{H}} + \boldsymbol{v}_t \right\} \\
&= \iiint \left(\mathbf{H}\boldsymbol{b}_t + \mathbf{\Delta}_t^{\mathbb{H}}\right) p(\mathbf{\Delta}_t^{\mathbb{H}}|\boldsymbol{b}_t, \mathcal{D})p(\boldsymbol{b}_{1:t}, \boldsymbol{m}_{1:t}|\widetilde{\boldsymbol{b}}_{1:t-1}, \mathcal{D})\mathrm{d}\mathbf{\Delta}_t^{\mathbb{H}}\mathrm{d}\boldsymbol{m}_t\mathrm{d}\boldsymbol{b}_t \\
&= \mathbf{H}\hat{\boldsymbol{b}}_t' + \hat{\mathbf{\Delta}}_t^{\mathbb{H}}.
\end{aligned} \tag{34}$$

The measurement prediction covariance is calculated as:

$$\begin{aligned}
\mathbf{P}_t^{\tilde{b}\tilde{b}} &= \mathbb{E}_{p(\widetilde{\boldsymbol{b}}_t|\widetilde{\boldsymbol{b}}_{1:t-1}, \mathcal{D})}\left\{ \left(\widetilde{\boldsymbol{b}}_t - \hat{\boldsymbol{b}}_t'\right)\left(\widetilde{\boldsymbol{b}}_t - \hat{\boldsymbol{b}}_t'\right)^\top \right\} \\
&= \iiint \left(\mathbf{H}\boldsymbol{b}_t + \mathbf{\Delta}_t^{\mathbb{H}} + \boldsymbol{v}_t - \widetilde{\boldsymbol{b}}_t'\right)\left(\mathbf{H}\boldsymbol{b}_t + \mathbf{\Delta}_t^{\mathbb{H}} + \boldsymbol{v}_t - \widetilde{\boldsymbol{b}}_t'\right)^\top P_t^2 \mathrm{d}\mathbf{\Delta}_t^{\mathbb{H}}\mathrm{d}\boldsymbol{m}_t\mathrm{d}\boldsymbol{b}_t \\
&= \mathbf{H}\mathbf{P}_t'\mathbf{H}^\top + \mathbf{P}_t^{\mathbb{H}} + \mathbf{R}_t,
\end{aligned} \tag{35}$$

where $P_t^2 = p(\mathbf{\Delta}_t^{\mathbb{H}}|\boldsymbol{b}_t, \mathcal{D})p(\boldsymbol{b}_{1:t}, \boldsymbol{m}_{1:t}|\widetilde{\boldsymbol{b}}_{1:t-1}, \mathcal{D})$.

And the mutual covariance of the state prediction and the measurement prediction is calculated as:

$$\begin{aligned}
\mathbf{P}_t^{b\tilde{b}} &= \mathbb{E}_{p(\widetilde{\boldsymbol{b}}_t|\widetilde{\boldsymbol{b}}_{1:t-1}, \mathcal{D})}\left\{ \left(\boldsymbol{b}_t - \hat{\boldsymbol{b}}_t'\right)\left(\widetilde{\boldsymbol{b}}_t - \hat{\boldsymbol{b}}_t'\right)^\top \right\} \\
&= \iiint \left(\mathbf{F}\boldsymbol{b}_{t-1} + \mathbf{\Delta}_t^{\mathbb{F}} + \boldsymbol{w}_t - \hat{\boldsymbol{b}}_t'\right)\left(\mathbf{H}\boldsymbol{b}_t + \mathbf{\Delta}_t^{\mathbb{H}} + \boldsymbol{v}_t - \widetilde{\boldsymbol{b}}_t'\right)^\top P_t^2 \mathrm{d}\mathbf{\Delta}_t^{\mathbb{H}}\mathrm{d}\boldsymbol{m}_t\mathrm{d}\boldsymbol{b}_t \\
&= \mathbf{P}_t'\mathbf{H}^\top.
\end{aligned} \tag{36}$$

According to the Bayesian rule in Eq. 19, the posterior can be equivalent to:

$$p(\boldsymbol{b}_{1:t}, \boldsymbol{m}_{1:t}|\widetilde{\boldsymbol{b}}_{1:t}, \mathcal{D}) = \frac{p(\boldsymbol{b}_{1:t}, \boldsymbol{m}_{1:t}|\widetilde{\boldsymbol{b}}_{1:t-1}, \mathcal{D})}{p(\widetilde{\boldsymbol{b}}_t|\widetilde{\boldsymbol{b}}_{1:t-1}, \mathcal{D})} \tag{37}$$

Due to the self-conjugate property of Gaussian distributions under Bayesian theorem, the joint distribution of the state prediction and the measurement prediction is also Gaussian and can be expressed as follows:

$$p(\boldsymbol{b}_{1:t}, \widetilde{\boldsymbol{b}}_{1:t} | \widetilde{\boldsymbol{b}}_{1:t-1}, \mathcal{D}) = \mathcal{N}\left[\begin{pmatrix} \hat{\boldsymbol{b}}'_t \\ \widetilde{\boldsymbol{b}}'_t \end{pmatrix}, \begin{pmatrix} \mathbf{P}'_t & \mathbf{P}^{b\tilde{b}}_t \\ \left(\mathbf{P}^{b\tilde{b}}_t\right)^\top & \mathbf{P}^{\tilde{b}\tilde{b}}_t \end{pmatrix}\right], \tag{38}$$

Subsequently, we substitute Eq. 38 into Eq. 37 to obtain updates of the state and covariance as follows:

$$\hat{\boldsymbol{b}}_t = \hat{\boldsymbol{b}}'_t + \mathbf{P}^{b\tilde{b}}_t \left(\mathbf{P}^{\tilde{b}\tilde{b}}_t\right)^{-1} \left(\widetilde{\boldsymbol{b}}_t - \hat{\boldsymbol{b}}'_t\right), \tag{39}$$

$$\mathbf{P}_t = \mathbf{P}'_t - \mathbf{P}^{b\tilde{b}}_t \left(\mathbf{P}^{\tilde{b}\tilde{b}}_t\right)^{-1} \left(\mathbf{P}^{b\tilde{b}}_t\right)^\top. \tag{40}$$

Finally, if we define $\mathbf{P}^{b\tilde{b}}_t (\mathbf{P}^{\tilde{b}\tilde{b}}_t)^{-1}$ as $\mathbf{K}_t$ (so called Kalman gain), then Eqs. 39 and 40 can be expressed as:

$$\hat{\boldsymbol{b}}_t = \hat{\boldsymbol{b}}'_t + \mathbf{K}_t(\widetilde{\boldsymbol{b}}_t - \mathbf{H}\hat{\boldsymbol{b}}'_t - \hat{\boldsymbol{\Delta}}^{\mathbb{H}}_t), \tag{41}$$

$$\mathbf{P}_t = (\mathbf{I} - \mathbf{K}_t\mathbf{H})\mathbf{P}'_t. \tag{42}$$

## C  DETAILED TRAINING PROCEDURE FOR MEKF

The MeKF requires detection boxes as input during inference to produce an estimate of the target's state. However, existing MOT datasets typically only provide ground truth trajectories, which is insufficient for our end-to-end training pipeline. To address this, we construct paired sequences of detection boxes and ground truth trajectories.

Specifically, we first employ the YOLOX detector, pre-trained as described in Section 4.2, to generate a sequence of detections for each frame, ensuring consistency with the actual tracking process. At time $t$, the detector generates a set of $N_t$ detection boxes from a single frame, namely, $\mathcal{A}_t = \{\widetilde{\boldsymbol{b}}^n_t\}_{n=1,2,\dots,N_t}$, where $n$ stands for the index of the detection box. Subsequently, we match these detections to the ground truth (a set of $M_t$ boxes at time $t$, namely, $\mathcal{B}_t = \{\bar{\boldsymbol{b}}^m_t\}_{m=1,2,\dots,M_t}$) based on a standard IoU threshold of 0.8. This process can be formulated as:

$$\pi_t(m) = \begin{cases} \arg\max_n \text{IoU}(\bar{\boldsymbol{b}}^m_t, \widetilde{\boldsymbol{b}}^n_t), & \text{if } \text{IoU}(\bar{\boldsymbol{b}}^m_t, \widetilde{\boldsymbol{b}}^n_t) > 0.8, \\ 0, & \text{otherwise}, \end{cases} \tag{43}$$

where $\pi_t(m)$ defines the mapping from a ground truth box to a detection box. Specifically, $\pi_t(m) = n$ indicates that the $m$-th ground-truth box is successfully associated with the $n$-th detection. A value of $\pi_t(m) = 0$ signifies a matching failure, meaning the ground truth box remains unmatched, which often corresponds to a missed detection.

Based on Eq. 43, The matching follows these criteria:

- Each ground truth box is matched with at most one detection; if multiple detections surpass the IoU threshold, the one with the highest IoU is selected.
- A single detection can be associated with multiple ground truth boxes.

Following this matching procedure, we obtain a set of pair-wise tuples, each containing a ground truth box and its matched detection for a single target in a given frame, namely, $\mathcal{C}_t = \{\bar{\boldsymbol{b}}^m_t, \widetilde{\boldsymbol{b}}^{\pi_t(m)}_t\}$. Since our LSTM-based MeKF requires fixed-length sequences for training, we generate these by applying a sliding window of length $T$ (as defined in Eq. 13) to the full trajectories. Each resulting training sequence for a single target trajectory, generated from one sliding window, can be represented as $\mathbf{C} = [\mathcal{C}_1, \mathcal{C}_2, \dots, \mathcal{C}_T]$. The final training dataset, which we denote as $\mathcal{D}$, is the collection of all such sequences generated from all target trajectories. This dataset is then used to train the MeKF.

It should be noted that the IoU-based matching between detections and ground truth boxes is not always successful. Matching failures can occur, for instance, in cases of missed detections (i.e.,

no detection box is generated) or when a detection significantly deviates from its corresponding ground-truth box. In such scenarios where a match is lossed, we set $\widetilde{\boldsymbol{b}}_t = \mathbf{H}\hat{\boldsymbol{b}}'_t + \hat{\boldsymbol{\Delta}}_t^{\mathbb{H}}$ in Eq. 11. This configuration prompts the filter to perform only the state prediction for the current time step, and bypassing the measurement update process.

# D SUPPLEMENTARY EXPERIMENTS

## D.1 BENCHMARK RESULTS ON MOT17 AND MOT20

We incorporated full evaluations on the MOT17 and MOT20 benchmarks, as shown in Tables 7 and 8. These benchmarks are characterized by pedestrian tracking scenarios where motion patterns are predominantly linear and predictable (Hu et al., 2024). We retained the settings from DanceTrack and SportsMOT, modifying only the velocity thresholds $\Theta_{center}$ and $\Theta_{height}$ according to the normalized velocity distribution from training set (i.e. 0.0208 and 0.0001 for MOT17, 0.0209 and 0.0001 for MOT20).

Table 7: Performance comparison with SOTA methods on the MOT17 test set. The best/second results are shown in **bold**/underlined.

| Methods | HOTA ↑ | AssA↑ | IDF1↑ | DetA↑ | MOTA ↑ | IDs ↓ | FP $(10^4)$↓ | FN $(10^4)$↓ |
|---|---|---|---|---|---|---|---|---|
| ByteTrack (Zhang et al., 2022) | 63.1 | 62.0 | 77.3 | 64.5 | 80.3 | 2196 | 2.55 | **8.37** |
| OC-SORT (Cao et al., 2023) | 63.2 | 63.2 | 77.5 | - | 78.0 | 1950 | **1.51** | 10.80 |
| C-BIoU (Fan et al., 2023) | **64.1** | 63.7 | **79.7** | **64.8** | **81.1** | - | - | - |
| BUSCA (Vaquero et al., 2024) | 63.9 | 64.2 | 79.2 | 63.9 | 78.6 | **1428** | 2.46 | 9.45 |
| TOPICTrack (Cao et al., 2025) | 63.9 | 64.3 | 78.6 | 63.7 | 78.8 | 1515 | 1.70 | 10.11 |
| **MeMoSORT** (ours) | 63.9 | **64.5** | 79.3 | 63.6 | 78.7 | 2058 | 2.02 | 9.77 |

Table 8: Performance comparison with SOTA methods on the MOT20 test set. The best/second results are shown in **bold**/underlined.

| Methods | HOTA ↑ | AssA ↑ | IDF1 ↑ | DetA ↑ | MOTA ↑ | IDs ↓ | FP $(10^4)$↓ | FN $(10^4)$↓ |
|---|---|---|---|---|---|---|---|---|
| ByteTrack (Zhang et al., 2022) | 61.3 | 59.6 | 75.2 | 63.4 | 77.8 | 1223 | 2.62 | 8.76 |
| OC-SORT (Cao et al., 2023) | 62.1 | 62.0 | 75.9 | - | 75.5 | 913 | 1.80 | 10.80 |
| BPMTrack (Gao et al., 2024) | 62.3 | 60.9 | 76.7 | **63.9** | **78.3** | 1314 | 2.86 | **8.25** |
| BUSCA (Vaquero et al., 2024) | 61.8 | 63.5 | 76.3 | 60.3 | 72.7 | 1006 | 1.38 | 12.63 |
| TOPICTrack (Cao et al., 2025) | **62.6** | **65.4** | **77.6** | 60.0 | 72.4 | **869** | **1.10** | 13.11 |
| **MeMoSORT** (ours) | 61.9 | 63.8 | 75.7 | 60.2 | 72.5 | 1200 | 1.26 | 12.83 |

The results in Tables 7 and 8 indicate MeMoSORT achieves results comparable to established baselines on standard benchmarks. Notably, on MOT17, our method achieves the best AssA score, outperforming leading baselines like TOPICTrack and BUSCA, while maintaining a competitive HOTA score.

Our MeMoSORT shows significant performance gains on DanceTrack and SportsMOT, but only modest improvements on MOT17 and MOT20, revealing a performance discrepancy across datasets. We attribute this discrepancy on MOT17/20 to their distinct motion patterns and occlusion characteristics, which interact differently with our MeKF and Mo-IoU modules.

At first, pedestrians in MOT17/20 typically move in a linear and stable manner, making standard KFs based on the Markovian assumption sufficient for motion prediction. Conversely, targets in DanceTrack and SportsMOT exhibit long-term dependencies, a complexity that our MeKF is explicitly designed to handle. Consequently, the distinct advantage of our MeKF in modeling complex motion is not fully exploited on MOT17/20, resulting in performance marginally superior to standard KF-based methods.

Secondly, the motion characteristics of targets during occlusion in MOT17/20 significantly differ from those in DanceTrack and SportsMOT. For example, targets in the latter datasets often interact with complex and rapid movements during occlusion, whereas occluded pedestrians in MOT17/20 typically remain in a slow-motion. When targets within the surveillance area move slowly, the dynamic adaptability of Mo-IoU to varying speeds yields only limited gains.

Overall, despite these limiting factors on MOT17/20, MeMoSORT still outperforms several trackers that do not utilize NNs. This demonstrates that MeKF and Mo-IoU effectively solve the complex MOT problem without overfitting to the simplified scenarios in traditional benchmarks.

## D.2 Generality on Other Baseline Trackers

We applied the key components of MeMoSORT on other representative TBD trackers as baselines, including SORT, BoT-SORT and DeepSORT. They utilize KF as state estimation methods, while applying different association strategies in consideration of spatial and appearance information. From Table 9, significant improvements can be observed from all these baseline trackers after applying MeKF or Mo-IoU, demonstrating the generality of the proposed key components.

Table 9: Generality experiments of applying MeKF and Mo-IoU to other baseline trackers on the DanceTrack validation set.

| Baseline tracker | MeKF | Mo-IoU | HOTA ↑ | AssA ↑ | IDF1 ↑ | DetA ↑ | MOTA ↑ |
|---|---|---|---|---|---|---|---|
| BoT-SORT (Aharon et al., 2022) | | | 58.68 | 37.11 | 50.22 | 92.87 | 96.50 |
| | ✓ | | 68.28 | 50.72 | 66.40 | 91.97 | 97.40 |
| | | ✓ | 68.62 | 51.26 | 67.39 | 91.91 | 97.62 |
| SORT (Bewley et al., 2016) | | | 55.57 | 33.26 | 46.22 | 92.94 | 96.19 |
| | ✓ | | 63.64 | 43.48 | 56.55 | 93.21 | 96.95 |
| | | ✓ | 67.11 | 49.04 | 66.39 | 91.89 | 97.58 |
| DeepSORT (Wojke et al., 2017) | | | 53.68 | 31.02 | 44.14 | 92.97 | 95.98 |
| | ✓ | | 62.12 | 41.45 | 54.38 | 93.16 | 96.83 |
| | | ✓ | 64.18 | 44.15 | 57.31 | 93.36 | 97.07 |

# E Additional Experiments of MeKF

## E.1 Sensitivity Analysis of MeKF's Training Data Volume

We conduct experiments on the DanceTrack validation set to investigate the data effciency of MeMoSORT by varying the training data volume for MeKF. The results shown in Table 10 demonstrate remarkable data efficiency: even when the training data is restricted to only 5%, MeMoSORT maintains a robust HOTA score, exhibiting negligible degradation compared to the full-data baseline. This confirms that our MeKF possesses strong robustness, relying on the physical prior and the Bayesian framework as inductive bias to maintain high performance even when training samples are scarce.

Table 10: Performance comparison under varying training data volume on the DanceTrack validation set. The best/second results are shown in **bold**/underlined.

| Percentage of training data | HOTA ↑ | AssA ↑ | IDF1 ↑ | DetA ↑ | MOTA ↑ |
|---|---|---|---|---|---|
| 5% | 65.56 | 46.15 | 58.18 | 93.19 | 96.79 |
| 10% | 66.13 | 46.81 | 59.27 | **93.32** | 96.94 |
| 25% | 66.62 | 47.77 | 61.07 | 92.97 | 97.24 |
| 50% | 67.13 | 48.53 | 61.89 | 92.91 | 97.26 |
| 100% | **67.41** | **49.58** | **66.41** | 91.69 | **97.55** |

## E.2 Sensitivity Analysis of MeKF's Memory Dimension

Table 11: Sensitivity analysis of MeKF's memory dimension on the DanceTrack validation set. The best/second results are shown in **bold**/underlined.

| Dimension | HOTA ↑ | AssA ↑ | IDF1 ↑ | DetA ↑ | MOTA ↑ |
|---|---|---|---|---|---|
| 8 | 60.47 | 39.32 | 52.34 | 93.07 | 96.68 |
| 16 | 63.67 | 43.55 | 56.81 | **93.10** | 96.92 |
| 32 | **67.41** | **49.58** | **66.41** | 91.69 | 97.55 |
| 64 | 67.11 | 49.04 | 66.39 | 91.89 | **97.58** |

As shown in Table 11, we analyze the sensitivity of MeKF's memory dimension. HOTA, AssA and IDF1 achieve their highest values at the dimension of 32. However, further increasing the dimension to 64 leads to a slight degradation in performance. This trend suggests that choosing 32 as the dimension of memory provides an optimal trade-off, offering sufficient capacity to model complex motions without introducing overfitting.

### E.3 GENERALIZATION EXPERIMENTS OF MEKF

To assess the generalization capability of MeKF, we conduct a cross-dataset evaluation on Dance-Track and SportsMOT. The experiments focus on training MeKF on one dataset's training set and testing it on the other dataset's validation set, with the results detailed in Table 12.

As expected, MeKF achieves its best performance when trained and tested on the same dataset, with only a slight degradation observed in cross-dataset experiments. The minimal performance gap in these experiments validate that MeKF learns robust and transferable motion patterns, highlighting its strong generalization capability.

Table 12: Generalization experiments of MeKF on DanceTrack and SportsMOT

| Training Dataset | Testing Dataset | HOTA ↑ | AssA ↑ | IDF1 ↑ | DetA ↑ | MOTA ↑ |
|---|---|---|---|---|---|---|
| DanceTrack | DanceTrack | 67.41 | 49.58 | 66.41 | 91.69 | 97.55 |
| SportsMOT | DanceTrack | 65.83 | 46.53 | 59.93 | 93.20 | 97.21 |
| SportsMOT | SportsMOT | 79.77 | 68.18 | 78.84 | 93.35 | 98.43 |
| DanceTrack | SportsMOT | 78.70 | 66.57 | 77.80 | 93.09 | 97.79 |

## F  ADDITIONAL EXPERIMENTS OF MO-IOU

### F.1  DISCUSSION ON THE DESIGN OF MAT FORMULATIONS

We introduce two continuous baselines based on $\tanh(\cdot)$ and $\mathrm{sigmoid}(\cdot)$ functions to evaluate robustness against velocity noise. This comparison aims to verify whether continuous mappings propagate minor jitters compared to the stable discrete design. The formulations are defined as follows:

$\tanh(\cdot)$ function form:

$$p_t = \frac{1 + \tanh(\alpha(\dot{c}_{t-1} - \Theta_{\text{center}}))}{2}(M_{\text{fast}} - M_{\text{slow}}) + M_{\text{slow}}, \tag{44}$$

$$q_t = \frac{1 + \tanh(\alpha(\dot{h}_{t-1} - \Theta_{\text{height}}))}{2}(N_{\text{fast}} - N_{\text{slow}}) + N_{\text{slow}}, \tag{45}$$

$\mathrm{sigmoid}(\cdot)$ function form:

$$p_t = \mathrm{sigmoid}\big(\alpha(\dot{c}_{t-1} - \Theta_{\text{center}})\big)(M_{\text{fast}} - M_{\text{slow}}) + M_{\text{slow}}, \tag{46}$$

$$q_t = \mathrm{sigmoid}\big(\alpha(\dot{h}_{t-1} - \Theta_{\text{height}})\big)(N_{\text{fast}} - N_{\text{slow}}) + N_{\text{slow}}, \tag{47}$$

where $\alpha$ is a scalar controlling the transition rate of the functions. Specific results are presented in the Table 13.

Table 13: Sensitivity analysis of continuous MAT functions on DanceTrack validation set. Values represent HOTA scores, with deviations from the original binary form in parentheses.

| $\alpha$ | 1 | 10 | $10^2$ | $10^3$ | $10^4$ | $10^5$ |
|---|---|---|---|---|---|---|
| $\tanh(\cdot)$ | 76.96 (-0.58) | 77.16 (-0.38) | 77.21 (-0.33) | 77.47 (-0.07) | **77.54 (-0.00)** | **77.54 (-0.00)** |
| $\mathrm{sigmoid}(\cdot)$ | 76.91 (-0.63) | 77.25 (-0.29) | 76.94 (-0.60) | 77.27 (-0.27) | **77.54 (-0.00)** | **77.54 (-0.00)** |

As illustrated in Table 13, the HOTA score exhibits a consistent upward trend as the scaling factor $\alpha$ increases. The performance reaches its optimum when $\alpha$ is large ($10^4 \sim 10^5$), at which point

the continuous functions approximate the original discrete binary form. Conversely, smoother transitions (smaller $\alpha$) lead to performance degradation. This empirical evidence validates that discrete binary switching is superior to continuous tuning for this application.

Overall, the discrete design functions as a parameter quantizer, which forces all targets within a binary speed level ("fast" or "slow") to utilize the same association parameters ($p_t$ and $q_t$), ignoring minor speed fluctuations. This parameter consistency yields a stable and uniform association cost matrix, which is critical for the Hungarian algorithm to find global optima without flickering.

## F.2 SENSITIVITY ANALYSIS OF MO-IOU'S THRESHOLDS

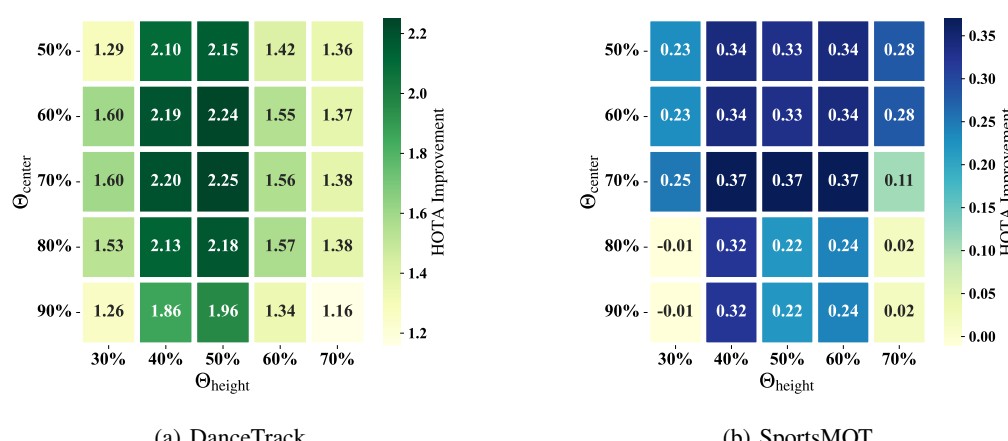

(a) DanceTrack

(b) SportsMOT

Figure 5: Sensitivity analysis of Mo-IoU's thresholds, $\Theta_{\text{height}}$ and $\Theta_{\text{center}}$, on the (a) DanceTrack validation set and (b) SportsMOT validation set. The heatmap displays the HOTA improvement (in points) relative to fixed parameters. The analysis reveals a peak performance gain at the configuration of $\Theta_{\text{height}}$=50% and $\Theta_{\text{center}}$=70%. The broad area of significant improvement demonstrate the robustness of our proposed Motion-Adaptive Technique (MAT) to hyperparameter variations.

To evaluate the sensitivity on the threshold of our proposed Mo-IoU, we conduct an analysis on $\Theta_{\text{height}}$ and $\Theta_{\text{center}}$. As depicted in Figure 5, we explore various parameter combinations and report the resulting HOTA improvement over the static parameter setting. The values for both thresholds are determined based on the percentile of the target speed distribution observed in the training set; for instance, a 50% setting corresponds to the median speed.

The results indicate that the optimal configuration ($\Theta_{\text{height}}$=50%, $\Theta_{\text{center}}$=70%) achieves a peak HOTA gain on both datasets. More importantly, the heatmap reveals a large contiguous region where performance gains consistently exceed the fixed parameter setting. This demonstrates that Mo-IoU is not highly sensitive to the precise choice of thresholds, validating the robustness and practical applicability of MAT.

## F.3 SENSITIVITY ANALYSIS OF MO-IOU'S PARAMETER

To evaluate the sensitivity on the threshold of our proposed Mo-IoU, we have added a sensitivity analysis for the MAT parameters ($M_{\text{slow}}$, $M_{\text{fast}}$, $N_{\text{slow}}$, and $N_{\text{fast}}$) to verify their impact on tracking performance. The results are summarized in Table 14.

As illustrated in the table, the MAT demonstrates a significant performance margin across diverse parameter configurations. Specifically, even under suboptimal settings where parameters significantly deviate from the optimal values (e.g., the most extreme case of $M_{\text{slow}}$=0.3, $M_{\text{fast}}$=0.4 combined with $N_{\text{slow}}$=4, $N_{\text{fast}}$=3), the HOTA score remains above 75. This wide operating range of the system's performance stems from the intrinsic physical consistency of the MAT, rather than relying on fine-tuned heuristics.

Table 14: Sensitivity analysis of MAT parameters on DanceTrack validation set. Values represent HOTA scores, with rows denote expansion parameters ($M_{slow}$, $M_{fast}$) and columns denote height parameters ($N_{slow}$, $N_{fast}$).

| HOTA ↑ | | $(N_{slow}, N_{fast})$ | | |
|---|---|---|---|---|
| | | (2,1) | (3,2) | (4,3) |
| ($M_{slow}$, $M_{fast}$) | (0.3,0.4) | 77.02 | 76.34 | 75.59 |
| | (0.4,0.5) | 77.51 | 76.46 | 75.77 |
| | (0.5,0.6) | **77.54** | 76.46 | 75.91 |
| | (0.6,0.7) | 77.48 | 76.59 | 75.85 |
| | (0.7,0.8) | 77.01 | 76.27 | 75.60 |

## G    CASE ANALYSIS

To provide an intuitive understanding of the tracking behavior, we present several representative cases that illustrate how the algorithms perform under challenging scenarios. These examples are selected from different sequences to highlight typical situations where identity preservation is difficult, such as temporary occlusions or group separation. By examining these cases, we aim to complement the quantitative results, offering a clearer picture of the strengths of our proposed MeMoSORT.

### G.1    CASE 1: OCCLUSION

We analyze a video segment from the DanceTrack dataset where two targets cross paths, leading to a temporary occlusion. As shown in Figure 6, each subfigure contains a tracking results plot (left) and representative frames (right). In the tracking results plot, each ground truth (GT) identity is shown as a vertical line, while colors denote tracking identities. Frames where the GT identity has been missed are left blank, and thick dots mark the temporal positions where ID switches occur. Dashed arrows connect the temporal positions in the tracking results plot to the corresponding frames. In the DiffMOT algorithm, the IDs of the two targets are swapped when encountering occlusion during crossing. In contrast, our proposed MeMoSORT successfully maintains consistent IDs throughout the occlusion, demonstrating its robustness in handling occlusions and interactions.

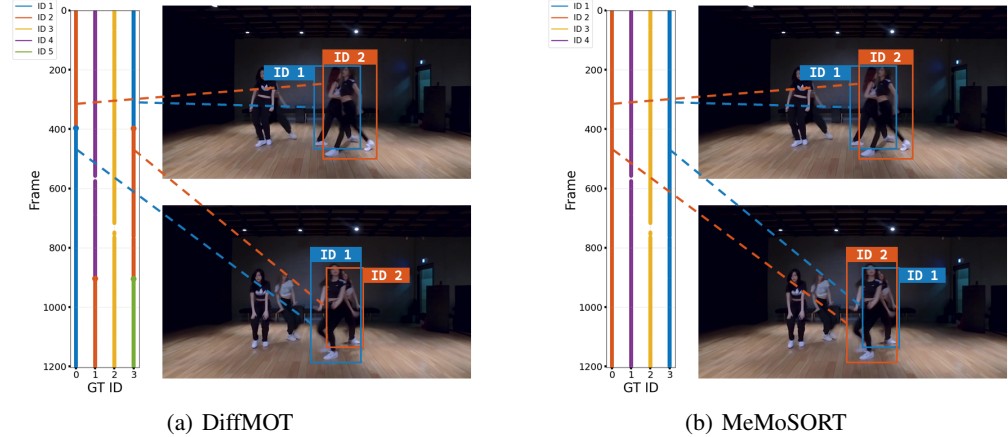

(a) DiffMOT                                          (b) MeMoSORT

Figure 6: Comparison of DiffMOT and MeMoSORT in a crossing scenario. (a) DiffMOT shows ID switch when two targets cross paths. (b) MeMoSORT preserves consistent IDs, demonstrating stronger robustness in handling interactions.

### G.2    CASE 2: GROUP SEPARATION

To further assess the robustness of the tracker, we examine a group separation scenario from the SportsMOT dataset. In this sequence, three targets move closely together, merging and separating, with frequent interactions and occlusions making identity tracking particularly challenging. As shown in Figure 7, DiffMOT fails to maintain ID consistency during separation, resulting in swapped IDs. In contrast, MeMoSORT effectively preserves stable IDs, showing its advantage in recovering from occlusion and maintaining robustness in group interaction scenarios.

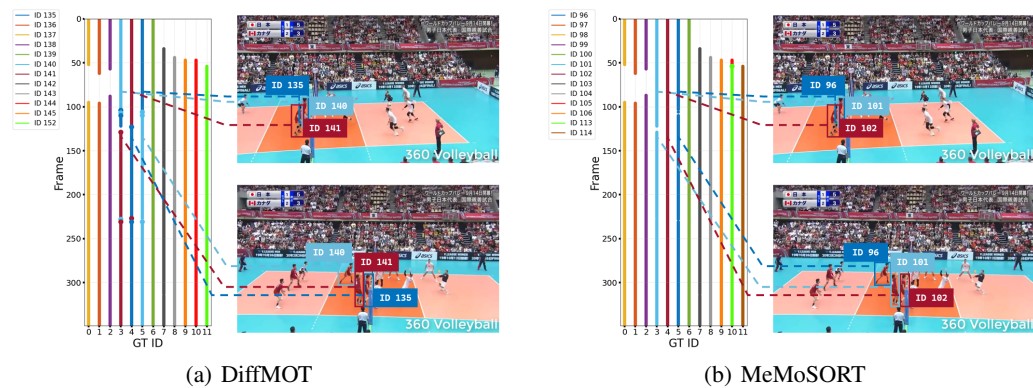

(a) DiffMOT                                        (b) MeMoSORT

Figure 7: Comparison of DiffMOT and MeMoSORT in a group separation scenario. (a) DiffMOT shows ID switch when three targets separate. (b) MeMoSORT preserves consistent IDs, demonstrating stronger robustness in handling group interactions and occlusion recovery.

### G.3    ADDITIONAL VISUALIZATIONS

Fig. 8 presents additional qualitative comparisons between our method and DiffMOT on the Dance-Track and SportsMOT validation sets. Similar to the sequences shown earlier, these cases highlight challenging scenarios such as frequent occlusions and complex interactions, where our approach demonstrates more stable identity preservation. These results further validate the effectiveness of our method under real-world challenges.

## H    THE USE OF LARGE LANGUAGE MODELS

During the preparation of this manuscript, a Large Language Model (LLM) was utilized to assist with language polishing, grammar correction, and improving overall readability. The LLM's role was strictly limited to editing and rephrasing. All intellectual content, including the core ideas, methodology, experiments, and conclusions, is the original work of the authors.

## I    REPRODUCIBILITY STATEMENT

To maintain the integrity of the double-blind review, our source code will be made available to the reviewers and area chairs via a private link during the official discussion period. We are committed to releasing our code publicly upon acceptance of the manuscript.

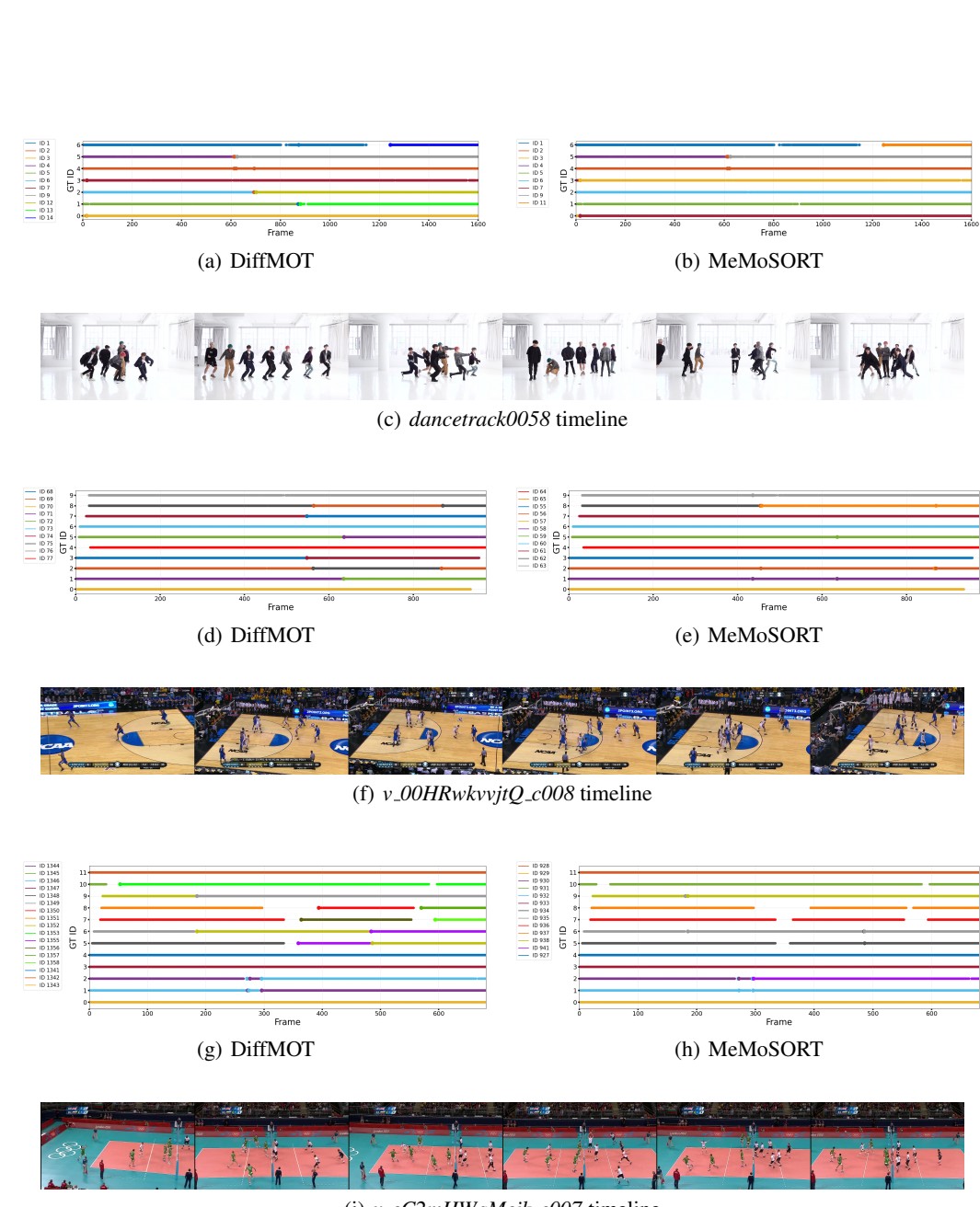

(a) DiffMOT

(b) MeMoSORT

(c) *dancetrack0058* timeline

(d) DiffMOT

(e) MeMoSORT

(f) *v_00HRwkvvjtQ_c008* timeline

(g) DiffMOT

(h) MeMoSORT

(i) *v_cC2mHWqMcjk_c007* timeline

Figure 8: Tracking results visualizations on supplementary videos from the DanceTrack and SportsMOT validation sets. (a-c) video *dancetrack0058* from DanceTrack. (d-f) video *v_00HRwkvvjtQ_c008* from SportsMOT. (g-i) video *v_cC2mHWqMcjk_c007* from SportsMOT.

