# OpenReview forum: "MeMoSORT: Memory-Assisted Filtering and Motion-Adaptive Association Metric for Multi-Person Tracking"
_ICLR.cc/2026/Conference — Submitted to ICLR 2026_

### Official Review · Reviewer_DhRY · 2025-10-27

**Soundness:** 2
**Presentation:** 3
**Contribution:** 2
**Rating:** 6
**Confidence:** 5

**Summary:**

This paper introduces MeMoSORT, a simple, online, and real-time Multi-Object Tracking (MOT) algorithm designed specifically for human-dominant scenarios characterized by complex motion and severe occlusions. The method addresses two fundamental limitations of conventional tracking-by-detection (TBD) methods: the motion mismatch inherent in the Kalman Filter (KF) and the rigidity of standard Intersection over Union (IoU) association. MeMoSORT proposes two primary innovations: the Memory-assisted Kalman Filter (MeKF), which integrates memory-augmented neural networks to explicitly compensate for the discrepancy between assumed linear dynamics and actual non-Markovian object motion and the Motion-adaptive IoU (Mo-IoU), a spatial association metric that adaptively expands the matching region and incorporates height similarity based on the target’s normalized speeds.

**Strengths:**

- MeMoSORT achieves SOTA performance across multiple metrics, notably HOTA, on DanceTrack and SportsMOT.
- Mo-IoU provides a significant advantage in handling severe occlusions. It jointly controls the expansion scale (EIoU) and height weighting (HIoU) using the Motion-Adaptive Technique (MAT), which uses normalized speeds to adapt parameters discretely. This adaptive parameter selection ensures more robust and accurate tracking than existing fixed-parameter IoU variants.
- The overall presentation of the paper is clear and easy to follow

**Weaknesses:**

- The entire motivation for the Memory-assisted Kalman Filter (MeKF) rests on overcoming the "fundamental limitations of the linear, first-order Markovian motion model". The authors specifically implement the standard KF using a constant velocity model for the state transition matrix F. While the paper visually demonstrates that complex human movements (e.g., phased switching, predictable back-and-forth patterns) violate the Markovian assumption, the justification for choosing the most simplistic linear prior as the failure point is weak.
- The Motion-Adaptive Technique (MAT) employs a discrete piecewise design to set Mo-IoU's parameters ($p_t, q_t$) for efficiency, switching based on predefined, fixed velocity thresholds ($\Theta_{\text{center}}$, $\Theta_{\text{height}}$). Relying on this abrupt, binary switching rather than continuous tuning may introduce instability or non-smooth changes in the association cost function when normalized speeds fluctuate near these critical boundary thresholds, potentially destabilizing tracking in the fast, variable motion characteristic of DanceTrack and SportsMOT.

**Questions:**

1. The MeKF training relies on a specific dataset generation process using matched YOLOX detections and ground truth. Could the authors provide a sensitivity analysis or further discussion on how the performance of MeKF might change if a lower-performing or different detector were used to generate the training dataset?
2. The MeMoSORT framework is specifically tailored for human-dominant scenarios characterized by complex, non-Markovian motion and severe occlusions (DanceTrack, SportsMOT). While the cross-dataset evaluation between DanceTrack and SportsMOT showed only a "slight degradation" in performance, validating the learning of transferable motion patterns, have the authors tested MeMoSORT's generalization capability on more conventional Multi-Object Tracking (MOT) benchmarks, such as **MOT17** or **MOT20**?
3. The MeKF is a hybrid filter designed to retain the stability of the classic Bayesian structure by explicitly compensating the physics-based prior (using $\Delta\hat{F}_t$, $\Delta\hat{H}_t$) rather than fully replacing it. The authors claim this approach "robustly ensures the stability of the state estimation" and provides a "failsafe" compared to purely data-driven methods (like DiffMOT or Mamba-based trackers). Can the authors elaborate on the theoretical or empirical mechanisms that prevent potential instability when the NN components, such as the covariance compensation terms $P_t^F$ and $P_t^H$, are trained and integrated into the Gaussian approximation framework, particularly when compared to other hybrid methods that discard the physics prior entirely?

---

> ### Author Response · Authors · 2025-12-03
> **Response to Weaknesses**
>
> **Note: References cited in this and subsequent responses are listed in the final comment.**
>
> * **Different motion model**
>   We thank the reviewer for this insightful comment. While we used the constant velocity (CV) model as a baseline, our core argument is that even more complex, physics-based motion models (e.g., constant acceleration (CA), coordinated turn (CT), constant turn rate and acceleration (CTRA)) based filters built upon them, are all fundamentally **limited by their mismatch with the non-Markovian dynamics** of real-world motion. To validate this, we have updated the manuscript to add comparsions in **Table 4** (please see **L 459-476** for details) that compare the standard KF using different motion models. The results are as follows:
>
>   **Table 1.** Selected performance comparison of different filter on the DanceTrack validation set. The best/second results are shown in **bold**/$\underline{\mathrm{underlined}}$.
>   |Filter|HOTA↑|AssA↑|IDF1↑|MOTA↑|DetA↑|
>   |:-|:-:|:-:|:-:|:-:|:-:|
>   |KF (CV)|56.94|34.92|48.18|96.35|**92.91**|
>   |KF (CA)|57.55|39.46|54.64|92.98|84.18|
>   |KF (CT)|57.32|39.21|54.25|93.08|84.05|
>   |KF (CTRA)|58.13|40.43|55.54|93.09|83.83|
>   |**MeKF** (ours)|**67.41**|**49.58**|**66.41**|**97.55**|$\underline{91.69}$|
>
>   This is because all these complex motion models still adhere to the first-order Markovian assumption (memoryless); that is, the state prediction at time $t$ is conditioned only on the state at $t-1$. This assumption is a fundamental mismatch with real-world complex human movements (e.g., a dancer consistently spinning after a specific jump), which exhibit long-term temporal dependencies. Consequently, the primary factor degrading tracking performance is this dynamic mismatch rather than the specific choice of a CV model. By explicitly addressing these non-Markovian dynamics, our MeKF significantly outperforms even the most complex Markovian filter (KF-CTRA) by a massive margin of **+9.28** HOTA.
>
> * **Design of MAT formulations**
>   Actually, the discrete design delivers more stable performance because it effectively **filters out noise inherent in inaccurate velocity estimates**. Conversely, a continuous and smooth mapping propagates minor velocity jitters and noise, causing constant, slight fluctuations in the association cost matrix, thereby inducing unnecessary instability. To validate this, we designed and tested two continuous parameter adjustment schemes based on $\tanh(\cdot)$ and $\text{sigmoid}(\cdot)$ functions, formulated as follows:
>   * $\tanh(\cdot)$ form:
>   $$
>   p_t=\frac{1+\tanh(\alpha(\dot{c}\_{t-1}-\Theta\_{\text{center}}))}{2}(M\_{\text{fast}}-M\_{\text{slow}})+M\_{\text{slow}},  \ \ \ \ \ \ \ \
>   q_t=\frac{1+\tanh(\alpha(\dot{h}\_{t-1}-\Theta\_{\text{height}}))}{2}(N\_{\text{fast}}-N\_{\text{slow}})+N\_{\text{slow}},
>   $$
>   * $\text{sigmoid}(\cdot)$ form:
>   $$
>   p_t=\text{sigmoid}\big(\alpha(\dot{c}\_{t-1}-\Theta\_{\text{center}})\big)(M\_{\text{fast}}-M\_{\text{slow}})+M\_{\text{slow}},  \ \ \ \ \ \ \ \
>   q_t=\text{sigmoid}\big(\alpha(\dot{h}\_{t-1}-\Theta\_{\text{height}})\big)(N\_{\text{fast}}-N\_{\text{slow}})+N\_{\text{slow}}
>   $$
>
>   where $\alpha$ is a scalar controlling the transition rate of the functions. We performed new ablation studies based on these formulations. The specific results are presented in the **Table 2** below (please refer to **Appendix F.1, L 1160-1195** in the revised manuscript for details).
>
>   **Table 2.** Sensitivity analysis of continuous MAT functions on DanceTrack validation set. Values represent HOTA scores, with deviations from the original binary form in parentheses.
>   |$\alpha$|$1$|$10$|$10^2$|$10^3$|$10^4$|$10^5$|
>   |:-:|:-:|:-:|:-:|:-:|:-:|:-:|
>   |$\text{tanh}(\cdot)$|76.96 (-0.58)|77.16 (-0.38)|77.21 (-0.33)|77.47 (-0.07)|**77.54 (-0.00)**|**77.54 (-0.00)**|
>   |$\text{sigmoid}(\cdot)$|76.91 (-0.63)|77.25 (-0.29)|76.94 (-0.60)|77.27 (-0.27)|**77.54 (-0.00)**|**77.54 (-0.00)**|
>
>   As illustrated in Table 2, the HOTA score exhibits a consistent upward trend as the scaling factor $\alpha$ increases. The performance reaches its optimum (77.54 HOTA) when $\alpha$ is large ($10^4$~$10^5$), at which point the continuous functions approximate the original discrete binary form. Conversely, smoother transitions (smaller $\alpha$) lead to performance degradation (-0.63 HOTA). This empirical evidence validates that discrete binary switching is superior to continuous tuning for this application.
>
>   Overall, the discrete design functions as a parameter quantizer, which forces all targets within a binary speed level (''fast'' or ''slow'') to utilize the same association parameters ($p_t$ and $q_t$), ignoring minor speed fluctuations. This parameter consistency yields a stable and uniform association cost matrix, which is critical for the Hungarian algorithm to find global optima without flickering.

---

> ### Author Response · Authors · 2025-12-03
> **Response to Question 1: Sensitivity Analysis on Detection Quality**
>
> We thank the reviewer for this insightful suggestion. In our original manuscript, all experiments and the MeKF training dataset were based on detections from the YOLOX-X model. We have added a new sensitivity analysis to **investigate the dependency of MeKF's training on the quality of the detection data**.
> Specifically, we replaced the detector with different sizes (YOLOX-S, -M, and -L) to generate distinct training datasets and retrained our MeKF accordingly. These specific experimental results have been added to the revised manuscript (please refer to **Table 5, L 478-496** for details). The detailed results are as follows:
>
> **Table 3.** Sensitivity analysis on detector dependency for MeKF training. The best/second results are shown in **bold**/$\underline{\mathrm{underlined}}$.
> | Detector | HOTA ↑ | AssA ↑ | IDF1 ↑ | MOTA ↑ | DetA ↑ |
> |:-|:-:|:-:|:-:|:-:|:-:|
> | YOLOX-S | 59.51 | 42.51 | 60.71 | 93.63 | 83.49 |
> | YOLOX-M | 63.22 | 44.85 | 62.34 | 96.31 | 89.20 |
> | YOLOX-L | $\underline{65.19}$ | $\underline{47.26}$ | $\underline{64.06}$ | $\underline{96.88}$ | $\underline{89.98}$ |
> | YOLOX-X | **67.41** | **49.58** | **66.41** | **97.55** | **91.69** |
>
> Notably, MeKF retains strong performance (59.51 HOTA) even with the lightweight YOLOX-S. Despite the significant drop in detection precision, the MeKF effectively learns to compensate for the higher observation bias. This demonstrates that our method is not strictly dependent on high-quality inputs; rather, it adaptively models the specific performance of the detector.

---

> ### Author Response · Authors · 2025-12-03
> **Response to Questions 2: Comparsions on MOT17 and 20**
>
> We have incorporated full evaluations on the MOT17 and MOT20 benchmarks into the revised manuscript (please refer to **Tables 7 and 8, L 1033-1082** for details). The experimental results demonstrate that our method maintains competitive performance on these challenging datasets, proving its robustness and broad applicability. The specific quantitative comparisons are presented in the tables below:
>
> **Table 4.** Performance comparison with SOTA methods on the MOT17 test set. The best/second results are shown in **bold**/$\underline{\mathrm{underlined}}$.
> |Methods|HOTA↑|AssA↑|IDF1↑|DetA↑|MOTA↑|IDs↓|FP($10^4$)↓|FN($10^4$)↓|
> |:-|:-:|:-:|:-:|:-:|:-:|:-:|:-:|:-:|
> |ByteTrack [1]|63.1|-|77.3|-|$\underline{\text{80.3}}$|2196|2.55|**8.37**|
> |OC-SORT [2]|63.2|63.2|77.5|-|78.0|1950|**1.51**|10.80|
> |C-BIoU [3]|**64.1**|63.7|**79.7**|**64.8**|**81.1**|-|-|-|
> |BUSCA [4]|$\underline{\text{63.9}}$|64.2|79.2|$\underline{\text{63.9}}$|78.6|**1428**|2.46|$\underline{\text{9.45}}$|
> |TOPICTrack [5]|$\underline{\text{63.9}}$|$\underline{\text{64.3}}$|78.6|63.7|78.8|$\underline{\text{1515}}$|$\underline{\text{1.70}}$|10.11|
> |**MeMoSORT** (ours)|$\underline{\text{63.9}}$|**64.5**|$\underline{\text{79.3}}$|63.6|78.7|2058|2.02|9.77|
>
> **Table 5.** Performance comparison with SOTA methods on the MOT20 test set. The best/second results are shown in **bold**/$\underline{\mathrm{underlined}}$.
> |Methods|HOTA↑|AssA↑|IDF1↑|DetA↑|MOTA↑|IDs↓|FP($10^4$)↓|FN($10^4$)↓|
> |:-|:-:|:-:|:-:|:-:|:-:|:-:|:-:|:-:|
> |ByteTrack [1]|61.3|-|75.2|-|$\underline{\text{77.8}}$|1223|2.62|$\underline{\text{8.76}}$|
> |OC-SORT [2]|62.1|62.0|75.9|-|75.5|$\underline{\text{913}}$|1.80|10.80|
> |BPMTrack [6]|$\underline{\text{62.3}}$|60.9|$\underline{\text{76.7}}$|**63.9**|**78.3**|1314|2.86|**8.25**|
> |BUSCA [4]|61.8|63.5|76.3|$\underline{\text{60.3}}$|72.7|1006|1.38|12.63|
> |TOPICTrack [5]|**62.6**|**65.4**|**77.6**|60.0|72.4|**869**|**1.10**|13.11|
> |**MeMoSORT** (ours)|61.9|$\underline{\text{63.8}}$|75.7|60.2|72.5|1200|$\underline{\text{1.26}}$|12.83|
>
> As shown in **Tables 4** and **5**, MeMoSORT achieves results close to SOTA methods like C-BIoU and TOPICTrack. Notably, on MOT17, our method achieves the best AssA of 64.5, outperforming recent baselines like TOPICTrack and BUSCA, and we rank the second HOTA. The slight variance in performance can be attributed to the characteristics of the datasets:
>
> * **Distinct motion patterns**: Pedestrians in MOT17/20 typically move in a linear and stable manner [7, 8], making standard KFs based on the Markovian assumption sufficient for motion prediction. Conversely, targets in DanceTrack and SportsMOT **exhibit long-term non-Markovian dynamics** (e.g., a dancer consistently spinning after a specific jump), a complexity that our MeKF is explicitly designed to handle. Consequently, the distinct advantage of our MeKF in modeling complex motion is not fully exploited on MOT17/20, resulting in performance marginally superior to standard KF-based methods.
>
> * **Distinct occlusion characteristics**: The motion characteristics of targets during occlusion in MOT17/20 significantly differ from those in DanceTrack and SportsMOT. For example, targets in the latter datasets often interact with complex and rapid movements during occlusion, whereas occluded pedestrians in MOT17/20 typically remain in a slow-motion. When targets within the surveillance area move slowly, the dynamic adaptability of Mo-IoU to varying speeds yields only limited gains.
>
> * **Robust Generalization**: Overall, despite these limiting factors on MOT17 and MOT20, MeMoSORT still outperforms several trackers that do not utilize neural networks. This demonstrates that MeKF and Mo-IoU effectively solve the complex MOT problem without overfitting to the simplified scenarios in traditional benchmarks.

---

> ### Author Response · Authors · 2025-12-03
> **Response to Questions 3**
>
> * **Mechanisms for Ensuring Stability in MeKF**
>   MeKF leverages the Bayesian framework to continuously correct prediction errors using detection data via the minimum MSE estimator [9, 10], substantially mitigating error accumulation compared to open-loop predictions. In extreme cases where the neural network's (NN) output is close to zero, MeKF naturally degenerates into a standard Kalman filter (KF) dominated by the physical model ($\mathbf{F}$, $\mathbf{H}$, $\mathbf{Q}_t$, and $\mathbf{R}_t$). This helps preserve stability and offers a safe fallback to the classic KF when NN contributions are small.
>
>   Regarding the covariance compensation terms ($\mathbf{P}\_t^\mathbb{F}$ and $\mathbf{P}_t^\mathbb{H}$), we construct them via factorization (e.g., $\mathbf{P}_t^\mathbb{F} = \mathbf{A}_t \mathbf{A}_t^\top $, where $\mathbf{A}_t$ is a vector regressed by a linear layer) rather than direct regression. This structurally guarantees SPD matrices, preventing numerical divergence. Furthermore, these terms are additive to the original covariance given by the standard KF (e.g., $\mathbf{P}'\_{t} = \mathbf{F} \mathbf{P}\_{t-1} \mathbf{F}^\top + \mathbf{Q}_t + \mathbf{P}^\mathbb{F}_t $), which can increase total uncertainty rather than replace the covariance directly. As a result, the filter automatically becomes more conservative (i.e., relies less on the NN corrections) when the predicted uncertainty is large, thereby avoiding instability caused by over-confident but inaccurate neural predictions.
>
> * **Comparison to Hybrid Methods that Discard the Physics Prior**
>   MeKF incorporates a physical model (e.g., a kinematic prediction model) as an inductive bias, tasking the NN solely with learning and compensating for the residual error. This residual learning paradigm simplifies the optimization objective: the model avoids learning physical laws from scratch, focusing only on deviations, which effectively enhances stability [11, 12]. Empirically, we observe stable training and inference behaviors across all benchmarks without covariance blow‑up or state divergence.
>
>   In contrast, hybrid methods that discard the physics prior must learn complete motion patterns entirely from data. This significantly increases the optimization degrees of freedom and dynamic uncertainty. Consequently, such models are more prone to instability, such as error amplification and state divergence, during recursive prediction [13].

---

> ### Author Response · Authors · 2025-12-03
> **References**
>
> [1] Y. Zhang, et al. "Bytetrack: Multi-object tracking by associating every detection box." (_ECCV 2022_)
> [2] J. Cao, et al. "Observation-centric sort: Rethinking sort for robust multi-object tracking." (_CVPR 2023_)
> [3] F. Yang, et al. "Hard to track objects with irregular motions and similar appearances? make it easier by buffering the matching space." (_WACV 2023_)
> [4] L. Vaquero, et al. "Lost and found: Overcoming detector failures in online multi-object tracking." (_ECCV 2024_)
> [5] X. Cao, et al. "TOPIC: a parallel association paradigm for multi-object tracking under complex motions and diverse scenes." (_IEEE TIP 2025_)
> [6] Y. Gao, et al. "BPMTrack: Multi-object tracking with detection box application pattern mining." (_IEEE TIP 2024_)
> [7] B. Hu, et al. "Trackssm: A general motion predictor by state-space model." arXiv preprint arXiv:2409.00487, 2024.
> [8] M. Adžemović, et al. "Engineering an Efficient Object Tracker for Non-Linear Motion." arXiv preprint arXiv:2407.00738 (2024).
> [9] S. Yan, et al. "Explainable gated Bayesian recurrent neural network for non-Markov state estimation." (_IEEE TSP 2024_)
> [10] K. Ito, et al. "Gaussian filters for nonlinear filtering problems." (_IEEE TAC 2002_)
> [11] A. Karpatne, et al. "Theory-guided data science: A new paradigm for scientific discovery from data." (_IEEE TKDE 2017_)
> [12] N. Shlezinger, et al. "Model-Based Deep Learning." (_Proceedings of the IEEE 2023_)
> [13] P. Lippe, et al. "Pde-refiner: Achieving accurate long rollouts with neural pde solvers." (_NIPS 2023_)

---

### Official Review · Reviewer_iKTz · 2025-10-31

**Soundness:** 3
**Presentation:** 2
**Contribution:** 2
**Rating:** 4
**Confidence:** 3

**Summary:**

This paper proposes MeMoSORT, a simple, online　MOT algorithm. First, the Memory-Assisted Kalman Filter (MeKF) uses LSTMs to correct discrepancies between the expected and actual object motion. Motion-adaptive IoU (Mo-IoU) reduces mismatches through an ad-hoc process that adaptively expands matching regions and incorporates height similarity. Several experiments demonstrate its effectiveness.

**Strengths:**

- This paper is easy to understand.
- The proposed method is easy to understand as it combines elementary techniques.

**Weaknesses:**

- The design of the proposed method is ad hoc
  - The design of the Expansion IoU and Height IoU techniques is ad hoc.
  - Furthermore, there is no hyperparameter study for the relevant parameters (M, N).

- Insufficient experiments
  - Generally, in tracking methods, comprehensive comparisons using multiple metrics like IDF1 and MOTA are common.
    - Specifically, metrics include IDF1, IDP, IDR, Recall, Precision, FP, FN, IDs, FM, MOTA, IDt, IDa, IDm, etc. This paper evaluates only a very limited subset of these metrics.
  - Furthermore, it does not compare with transformer-based methods like MOTR, MOTRv2, or their variants. It remains unclear whether the proposed method outperforms approaches like MOTRv2, especially on datasets such as DanceTrack.
  - Furthermore, the proposed method has not been compared against more general methods like MOT20 or MOT17.
  - Consequently, it is difficult to judge the effectiveness of the proposed method.

- Processing Speed
  - A strength of Detection by Track is speed. Does the proposed method have an advantage in computational speed compared to representative methods like Bytetrack?

- Concerns Regarding Domain Shift and Sparse Data
  - Since machine learning is used, performance degradation due to domain gaps between training and test data is a concern. This is thought to occur not only due to differences between datasets like dance track and sport mot, but also due to factors like differences in frame rate. Also, when training data is scarce, does the proposed method become inferior compared to the comparison methods?

**Questions:**

- Why wasn't it compared against more common methods like MOTA or IDF1?
- Why weren't more sophisticated transformer-based methods like MOTR or MOTRv2 used for comparison?
- Why weren't more common methods like MOT20 or MOT17 used for comparison?
- Does it inherit the weaknesses of machine learning compared to the comparison methods?
- Regarding computational speed, can it claim an advantage over Bytetrack?

---

> ### Comment · Reviewer_DhRY · 2025-11-12
>
> This reviewer might not be an expert nor famaliar with the MOT field. Here are some of my comments for this review.
>
> ---
> Weakness
> - Metrics like IDF1 and MOTA have been proofed to be suboptimal metrics for MOT. IDF1 overemphasizes on association and MOTA overemphasizes detection quality, this applies to other metrics mentioned by the reviewer. HOTA is now the main evaluation metrics for the MOT task due to its better balance between detection and association.
> - MOTRv2 has been regarded as a hard to reproduce work (or even not reproducible) by the community despite the opensource of its codebase [1]. Therefore I do not think it is fair for the author to compared with such work.
> - The overly focus on the FPS, as stated in my official comment.
> - With all due respect, I find the statement “Since machine learning is used, performance degradation due to domain gaps between training and test data is a concern…”  to be overly generic and lacking actionable insight. While domain shift is indeed a common challenge in machine learning, this comment is presented at a very high level and does not specify any concrete issue or evidence related to the proposed method in the MOT field.
>
> ---
> Questions
> - "Why wasn't it compared against more common methods like MOTA or IDF1?" These are not methods, they are metrics.
> - "Why weren't more common methods like MOT20 or MOT17 used for comparison?" These are benchmarks, not methods.
> - "Does it inherit the weaknesses of machine learning compared to the comparison methods?" I struggle to understand what the reviewer expect from the author rebuttal for this kind of question.
> - "Regarding computational speed, can it claim an advantage over Bytetrack?" As stated in my official comment. If everyone poses a constraint on the speed advantage, there will be no MOT paper after ByteTrack. I sincerely hope the reviewers reconsider the value of the work beyond runtime.
>
> ---
> Overall, I found this reviewer might not be very familiar with the field of MOT, and respectfully hope the AC can consider this context when making the final decision.
>
> [1] https://github.com/MCG-NJU/MeMOTR/issues/17

---

> > ### Comment · Reviewer_iKTz · 2025-11-22
> > **Comments for  the Authors**
> >
> > Dear Authors
> >
> > First and foremost, I would like to express my deepest apologies for having reviewed your paper, despite not being an expert in pattern recognition or computer vision.
> >
> > I received extremely valuable feedback from a reviewer possessing profound insight and expertise, highlighting the significant shortcomings in my own knowledge. Fortunately, this feedback enabled me to recognize my own substantial lack of expertise in computer vision and pattern recognition.
> >
> > Furthermore, I sincerely apologize again for the unexpected health issues that arose during most of the review period (approximately two weeks for five papers), which reduced the time I could dedicate to reviewing compared to my normal schedule.
> >
> > Considering these points, I have lowered the reliability rating from 3 to 1.
> >
> > ---
> >
> > If you have any comments or feedback regarding my review results, please don't hesitate to share them with me.
> > My comments may not be particularly insightful due to my significantly limited knowledge of pattern recognition and computer vision.
> >
> > In general, however, the discussion process itself, involving expert reviewers and knowledgeable authors, may yield valuable information for other reviewers and the AC regarding this review decision.
> >
> > I would be very pleased if this discussion period proves beneficial for both the authors and the reviewers.
> >
> > Thank you for your cooperation.

---

> ### Comment · Reviewer_iKTz · 2025-11-12
> **Comments for Reviewer DhRY, AC, and the Authors**
>
> I have revised the confidence rating downward (3->1) to reflect the comments below from the reviewer (DhRY). I would appreciate it if the author, AC, and other reviewers would consider the comments made to me by reviewer DhRY regarding my review.
>
> > - This reviewer might **not be an expert nor famaliar**...
> > - Overall, I found this reviewer might **not be** very **familiar** ..., hope the AC can **consider this context when making the final decision**.
>
> Also, typos have been corrected.

---

> ### Author Response · Authors · 2025-12-03
> **Response to Weaknesses: Principled Design and Hyperparameter Robustness**
>
> **Note: References cited in this and subsequent responses are listed in the final comment.**
>
> * **Design Rationale**
>   We respectfully clarify that the design of Mo-IoU is not ad hoc; rather, it is explicitly derived to address two fundamental failure modes in TBD-based tracking: **motion-induced spatial misalignment** and **occlusion-induced shape ambiguity**.
>     * **Expansion IoU (EIoU)**: Standard IoU often fails when targets undergo fast, non-Markovian motion, as the Kalman filter's prediction misaligns with the observation. Our EIoU is designed to compensate for this prediction error. By relaxing the box boundaries via the expansion factor $p_t$, we effectively enlarge the search space to capture reliable associations that would otherwise be missed by rigid IoU matching.
>     * **Height IoU (HIoU)**: Grounded in physical observation, HIoU exploits the fact that while width fluctuates under occlusion, height remains a robust geometric invariant. Functioning as a 1D-IoU vertical constraint, it distinguishes targets with similar horizontal overlaps but distinct vertical scales, significantly reducing ID switches.
>
> * **Hyperparameter Robustness**
>   Following your recommendation, we have added a sensitivity analysis for the MAT parameters ($M_{\text{slow}}$, $M_{\text{fast}}$, $N_{\text{slow}}$, and $N_{\text{fast}}$) to verify their impact on tracking performance. The results are summarized in **Table 1** below (please refer to **L 1242-1253** of the revised manuscript for full details):
>
>   **Table 1.** Sensitivity analysis of MAT parameters on DanceTrack validation set (results report HOTA). Rows denote expansion parameters ($M_{\text{slow}}$, $M_{\text{fast}}$), and columns denote height parameters ($N_{\text{slow}}$, $N_{\text{fast}}$).
>   | HOTA ↑ | $M_{\text{slow}}$=2, $M_{\text{fast}}$=1 | $M_{\text{slow}}$=3, $M_{\text{fast}}$=2 | $M_{\text{slow}}$=4, $M_{\text{fast}}$=3 |
>   |:-|:-:|:-:|:-:|
>   | $N_{\text{slow}}$=0.3, $N_{\text{fast}}$=0.4 | 77.02 | 76.34 | 75.59 |
>   | $N_{\text{slow}}$=0.4, $N_{\text{fast}}$=0.5 | 77.51 | 76.46 | 75.77 |
>   | $N_{\text{slow}}$=0.5, $N_{\text{fast}}$=0.6 | **77.54** | 76.46 | 75.91 |
>   | $N_{\text{slow}}$=0.6, $N_{\text{fast}}$=0.7 | 77.48 | 76.59 | 75.85 |
>   | $N_{\text{slow}}$=0.7, $N_{\text{fast}}$=0.8 | 77.01 | 76.27 | 75.60 |
>
>   As illustrated in the table, the MAT shows a significant performance margin across diverse parameter configurations. Specifically, even under suboptimal settings, the HOTA score remains **above 75**. This wide operating range directly addresses the concern of ad hoc design: it confirms that the system's performance **stems from the physical consistency of the MAT**, rather than relying on fine-tuned heuristics.

---

> ### Author Response · Authors · 2025-12-03
> **Response to Weaknesses: Clarification on Metrics and New Comparative Baselines**
>
> * **Clarification on Evaluation Metrics**
>   We respectfully clarify that **our original manuscript had included evaluation using both MOTA and IDF1. These metrics are already reported in Tables 1, 2, 3, 4, 5, 6, 7, and 8 of the original submission**.
>     * Following your suggestion, we have significantly expanded our evaluation to include detailed metrics such as **IDs, FP, and FN** in the comparative experiments on MOT17 and MOT20. Please refer to **Tables 3 and 4** provided later in this comment for the specific results.
>
> * **Comparison with Transformer-based Trackers**
>   We have added comparative experiments against MOTRv2 [1], MeMOTR [2], and MOTIP [3] on the DanceTrack test set. For the complete results, please refer to **Table 1 in our "Response to Weakness 1.1: Computational Cost and Training Efficiency" for Reviewer uXas**, as well as **Table 1, L 378-398** in the revised manuscript. Selected comparative results are presented below:
>
>     **Table 2**. Performance on DanceTrack test set, FPS on validation.
>     |Methods|HOTA↑|AssA↑|IDF1↑|DetA↑|MOTA↑|FPS↑|
>     |:-|:-:|:-:|:-:|:-:|:-:|:-:|
>     |**End-to-end tracker**|
>     |MOTRv2 [1]|69.9|64.4|76.0|83.7|92.1|10.2|
>     |MeMOTR [2]|63.4|52.3|65.5|77.0|85.4|17.7|
>     |MOTIP [3]|69.6|60.4|74.7|80.4|90.6|19.8|
>     |**Implicit memory-based filter**|
>     |**MeMoSORT** (Ours)|67.9|54.3|68.0|85.0|93.4|28.8|
>
>     The results in the table above demonstrate that MeMoSORT outperforms MeMOTR in accuracy (surpassing it by **+4.5 HOTA**). Furthermore, compared to transformer-based end-to-end trackers, our approach possesses **an advantage in terms of low computational overhead**, surpassing these methods by significant margins of **+18.6, +11.1, and +9.0 FPS**, respectively.

---

> ### Author Response · Authors · 2025-12-03
> **Response to Weakness: Additional Benchmarks on MOT17 and MOT20**
>
> We have incorporated full evaluations on the MOT17 and MOT20 benchmarks into the revised manuscript (please refer to **Tables 7 and 8, L 1033-1082** for details). The experimental results demonstrate that our method maintains competitive performance on these challenging datasets, proving its robustness and broad applicability. The specific quantitative comparisons are presented in the tables below:
>
> **Table 3.** Performance comparison with SOTA methods on the MOT17 test set. The best/second results are shown in **bold**/$\underline{\mathrm{underlined}}$.
> |Methods|HOTA↑|AssA↑|IDF1↑|DetA↑|MOTA↑|IDs↓|FP($10^4$)↓|FN($10^4$)↓|
> |:-|:-:|:-:|:-:|:-:|:-:|:-:|:-:|:-:|
> |ByteTrack [4]|63.1|-|77.3|-|$\underline{\text{80.3}}$|2196|2.55|**8.37**|
> |OC-SORT [5]|63.2|63.2|77.5|-|78.0|1950|**1.51**|10.80|
> |C-BIoU [6]|**64.1**|63.7|**79.7**|**64.8**|**81.1**|-|-|-|
> |BUSCA [7]|$\underline{\text{63.9}}$|64.2|79.2|$\underline{\text{63.9}}$|78.6|**1428**|2.46|$\underline{\text{9.45}}$|
> |TOPICTrack [8]|$\underline{\text{63.9}}$|$\underline{\text{64.3}}$|78.6|63.7|78.8|$\underline{\text{1515}}$|$\underline{\text{1.70}}$|10.11|
> |**MeMoSORT** (ours)|$\underline{\text{63.9}}$|**64.5**|$\underline{\text{79.3}}$|63.6|78.7|2058|2.02|9.77|
>
> **Table 4.** Performance comparison with SOTA methods on the MOT20 test set. The best/second results are shown in **bold**/$\underline{\mathrm{underlined}}$.
> |Methods|HOTA↑|AssA↑|IDF1↑|DetA↑|MOTA↑|IDs↓|FP($10^4$)↓|FN($10^4$)↓|
> |:-|:-:|:-:|:-:|:-:|:-:|:-:|:-:|:-:|
> |ByteTrack [4]|61.3|-|75.2|-|$\underline{\text{77.8}}$|1223|2.62|$\underline{\text{8.76}}$|
> |OC-SORT [5]|62.1|62.0|75.9|-|75.5|$\underline{\text{913}}$|1.80|10.80|
> |BPMTrack [9]|$\underline{\text{62.3}}$|60.9|$\underline{\text{76.7}}$|**63.9**|**78.3**|1314|2.86|**8.25**|
> |BUSCA [7]|61.8|63.5|76.3|$\underline{\text{60.3}}$|72.7|1006|1.38|12.63|
> |TOPICTrack [8]|**62.6**|**65.4**|**77.6**|60.0|72.4|**869**|**1.10**|13.11|
> |**MeMoSORT** (ours)|61.9|$\underline{\text{63.8}}$|75.7|60.2|72.5|1200|$\underline{\text{1.26}}$|12.83|
>
> As shown in **Tables 3** and **4**, MeMoSORT achieves results close to SOTA methods like C-BIoU and TOPICTrack. Notably, on MOT17, our method achieves the best AssA of 64.5, outperforming recent baselines like TOPICTrack and BUSCA, and we rank the second HOTA. The slight variance in performance can be attributed to the characteristics of the datasets:
>
> * **Distinct motion patterns**: Pedestrians in MOT17/20 typically move in a linear and stable manner [13, 14], making standard KFs based on the Markovian assumption sufficient for motion prediction. Conversely, targets in DanceTrack and SportsMOT **exhibit long-term non-Markovian dynamics** (e.g., a dancer consistently spinning after a specific jump), a complexity that our MeKF is explicitly designed to handle. Consequently, the distinct advantage of our MeKF in modeling complex motion is not fully exploited on MOT17/20, resulting in performance marginally superior to standard KF-based methods.
>
> * **Distinct occlusion characteristics**: The motion characteristics of targets during occlusion in MOT17/20 significantly differ from those in DanceTrack and SportsMOT. For example, targets in the latter datasets often interact with complex and rapid movements during occlusion, whereas occluded pedestrians in MOT17/20 typically remain in a slow-motion. When targets within the surveillance area move slowly, the dynamic adaptability of Mo-IoU to varying speeds yields only limited gains.
>
> * **Robust Generalization**: Overall, despite these limiting factors on MOT17 and MOT20, MeMoSORT still outperforms several trackers that do not utilize neural networks. This demonstrates that MeKF and Mo-IoU effectively solve the complex MOT problem without overfitting to the simplified scenarios in traditional benchmarks.

---

> ### Author Response · Authors · 2025-12-03
> **Response to Weaknesses: Computational Speed and Generalization Robustness**
>
> * **Advantage in Accuracy-Speed Trade-off**
>   We demonstrate that the substantial accuracy improvement of our method significantly outweighs the marginal reduction in speed. As evidenced in **Table 3** of the manuscript (**L 440-448**), even without ReID, MeKF outperforms the ByteTrack baseline by a massive margin (**+10.47 HOTA, +14.66 AssA, +18.23 IDF1**). Notably, these gains are realized at a highly competitive 60.8 FPS (vs. 74.5 FPS), confirming that our method retains real-time capability while effectively solving complex non-Markovian tracking problems.
>
> * **Robustness to Domain Shift, Frame Rate, and Data Sparsity**
>   * **Cross-Dataset Generalization**
>     We already included **cross-dataset** experiments in our **original manuscript's Appendix E.3 (L 1140-1156)** to validate MeMoSORT's robustness under domain shift. We conducted these generalization experiments using DanceTrack and SportsMOT as the respective source and target domains. The results show that MeMoSORT has strong generalization performance; for example, when training on DanceTrack and testing on SportsMOT, the HOTA score only dropped by **1.07** points compared to the performance when training directly on SportsMOT.
>
>   * **Adaptability to Frame Rates**
>     Actually, different frame rates **do not negatively impact** MeMoSORT's performance. Just like any standard KF-based tracker (e.g., ByteTrack), our method can be adapted to any video frame rate simply. Taking the MeKF's state prediction process as an example, the neural network components, including the LSTM for memory updates (Eq. 1) and the MLPs for generating compensation (Eqs. 2 and 3), are not functions of the time interval $\mathbf{\Delta}_t$. Instead, they depend only on the historical target states. Therefore, different frame rates do not affect the learned state estimation capability. When a video's frame rate changes, the only adjustment needed is to update the $\mathbf{\Delta}_t$ within the state transition matrix $\mathbf{F}$.
>
>     $$
>         \mathbf{m}_t = \mathcal{F}\_{\mathrm{LSTM}} (\mathbf{c}\_{t-1}, \mathbf{h}\_{t-1}, \mathbf{m}\_{t-1}), (1)
>     $$
>     $$
>         \hat{\mathbf{\Delta}}\^\mathbb{F}_t {=} \mathcal{F}\_{\mathrm{MLP}}\^1 (\mathbf{m}_t),  (2)
>     $$
>     $$
>         \hat{\mathbf{b}}'_t = \mathbf{F} \hat{\mathbf{b}}\_{t-1} + \hat{\mathbf{\Delta}}\^\mathbb{F}_t.  (3)
>     $$
>
>   * **Influence of Training Data Volume**
>
>     We added an analysis on the DanceTrack validation set to test MeMoSORT with less training data (please refer to **Table 5** below). By using a physical prior and the Bayesian framework, it maintains high performance even when training samples are scarce. Even with only **5%** of the training data, MeMoSORT achieves a HOTA score of **65.6**. This is very close to the score of 67.4 obtained with full data.
>
>     **Table 5.** Performance comparison under varying training data volume on the DanceTrack validation set. The best/second results are shown in **bold**/$\underline{\mathrm{underlined}}$.
>     | Percentage of training data | HOTA ↑ | DetA ↑ | AssA ↑ | MOTA ↑ | IDF1 ↑ |
>     |:-:|:-:|:-:|:-:|:-:|:-:|
>     | 5%                          | 65.56   | $\underline{93.19}$ | 46.15   | 96.79   | 58.18   |
>     | 10%                         | 66.13   | **93.32**  | 46.81   | 96.94   | 59.27   |
>     | 25%                         | 66.62   | 92.97   | 47.77   | 97.24   | 61.07   |
>     | 50%                         | $\underline{67.13}$ | 92.91   | $\underline{48.53}$ | $\underline{97.26}$ | $\underline{61.89}$ |
>     | 100%                        | **67.41** | 91.69   | **49.58** | **97.55** | **66.41** |

---

> ### Author Response · Authors · 2025-12-03
> **Response to Questions**
>
> * **More metircs**
>   We clarify that evaluation using both MOTA and IDF1 was **indeed included in our original manuscript**. These metrics were explicitly reported in **Tables 1 through 8** of the original submission
>
>   In addition to these traditional metrics, we also utilized HOTA, DetA, and AssA. We focused on these modern metrics (like HOTA) because they're more balanced. They measure both how well we find targets (detection) and how well we follow them (association) at the same time. This fixes the known biases of the older metrics: MOTA is mostly driven by detection accuracy, and IDF1 only cares about following targets, ignoring detection.
>   * Furthermore, following your suggestion, we have incorporated additional detailed metrics, such as **IDs, FP, and FN**. For specific results, please refer to our **Response to Weakness: Additional Benchmarks on MOT17 and MOT20**.
>
> * **Comparison with Transformer-based Trackers**
>   We have included comparisons with end-to-end transformer-based trackers, such as MOTIP, MeMOTR, and MOTRv2, in the revised manuscript. Please refer to **Table 1, L 378-398** of the revised manuscript for the detailed results. A summary of these results is also presented in **Table 2** of our **Response to Weaknesses: Clarification on Metrics and New Comparative Baselines**.
>
> * **More benchmark comparsions**
>   Following your suggestion, we have included experiments on the MOT17 and MOT20 datasets in the revised manuscript. In this new section, we provide a comprehensive evaluation of MeMoSORT’s performance compared to various other methods. Please refer to **Tables 3 and 4** of our **Response to Weakness: Additional Benchmarks on MOT17 and MOT20**.
>
> * **Mitigation of Common ML Weaknesses via Hybrid Design**
>   We clarify that while MeKF incorporates neural components, its design as **a neural network–aided Bayesian filter** effectively mitigates the common weaknesses (e.g., instability and poor generalization) associated with purely learning methods. Purely learning methods (e.g., DiffMOT's diffusion-based filter) can generate physically implausible predictions when the network produces outliers. In contrast, MeKF utilizes the physical motion model as a prior within the Baysian framework, with **the NN learning only a residual compensation**. This approach robustly ensures the stability of the state estimation [10, 11, 12].
>
> * **Advantage vs. ByteTrack: The Accuracy-Speed Trade-off**
>   Although MeKF introduces a small computational overhead, it delivers substantially higher accuracy than ByteTrack. As shown in **Table 3** of our manuscript (**L 440-448**), MeKF improves HOTA from 56.94 → 67.41 (**+10.47**), AssA from 34.92 → 49.58 (**+14.66**), and IDF1 from 48.18 → 66.41 (**+18.23**). Despite these significant gains, MeKF still runs at a competitive 60.8 FPS (vs. ByteTrack’s 74.5 FPS), maintaining real-time performance while offering far superior state estimation for complex non-linear motion.

---

> ### Author Response · Authors · 2025-12-03
> **References**
>
> [1] Y. Zhang, et al. "Motrv2: Bootstrapping end-to-end multi-object tracking by pretrained object detectors." (_CVPR 2023_)
> [2] R. Gao, et al. "MeMOTR: Long-term memory-augmented transformer for multi-object tracking." (_CVPR 2023_)
> [3] R. Gao, et al. "Multiple object tracking as id prediction." (_CVPR 2025_)
> [4] Y. Zhang, et al. "Bytetrack: Multi-object tracking by associating every detection box." (_ECCV 2022_)
> [5] J. Cao, et al. "Observation-centric sort: Rethinking sort for robust multi-object tracking." (_CVPR 2023_)
> [6] F. Yang, et al. "Hard to track objects with irregular motions and similar appearances? make it easier by buffering the matching space." (_WACV 2023_)
> [7] L. Vaquero, et al. "Lost and found: Overcoming detector failures in online multi-object tracking." (_ECCV 2024_)
> [8] X. Cao, et al. "TOPIC: a parallel association paradigm for multi-object tracking under complex motions and diverse scenes." (_IEEE TIP 2025_)
> [9] Y. Gao, et al. "BPMTrack: Multi-object tracking with detection box application pattern mining." (_IEEE TIP 2024_)
> [10] A. Karpatne, et al. "Theory-guided data science: A new paradigm for scientific discovery from data." (_IEEE TKDE 2017_)
> [11] N. Shlezinger, et al. "Model-Based Deep Learning." (_Proceedings of the IEEE 2023_)
> [12] S. Yan, et al. "Explainable gated Bayesian recurrent neural network for non-Markov state estimation." (_IEEE TSP 2024_)
> [13] B. Hu, et al. "Trackssm: A general motion predictor by state-space model." arXiv preprint arXiv:2409.00487, 2024.
> [14] M. Adžemović, et al. "Engineering an Efficient Object Tracker for Non-Linear Motion." arXiv preprint arXiv:2407.00738 (2024).

---

### Official Review · Reviewer_uXas · 2025-11-02

**Soundness:** 2
**Presentation:** 2
**Contribution:** 2
**Rating:** 2
**Confidence:** 5

**Summary:**

This paper introduces MeMoSORT, a method designed to address the problem of multiple-object tracking (MOT). It presents two main innovations: (a) a Memory-assisted Kalman Filter (MeKF), which employs a memory-augmented neural network (LSTM-based) to bridge the gap between assumed and actual motion patterns; and (b) a Motion-adaptive IoU (MoIoU), which dynamically expands the matching region and integrates height similarity to reduce association errors.

**Strengths:**

This paper introduces MeMoSORT, a method designed to address the problem of multiple-object tracking (MOT). It presents two main innovations: (a) a Memory-assisted Kalman Filter (MeKF), which employs a memory-augmented neural network (LSTM-based) to bridge the gap between assumed and actual motion patterns; and (b) a Motion-adaptive IoU (MoIoU), which dynamically expands the matching region and integrates height similarity to reduce association errors.  It shows SoTA results in  the  DanceTrack dataset.

**Weaknesses:**

There are several concerns regarding both novelty and practicality.  First, the MeKF component is computationally expensive and difficult to train due to its reliance on LSTMs. Consequently, the paper omits evaluation on more challenging datasets such as MOT20, which significantly limits the strength of the experimental validation. Second, using height as a discriminative feature for association is questionable. Estimating reliable person height from uncalibrated cameras is inherently difficult, and individuals with similar heights cannot be easily distinguished. Therefore, this feature does not effectively address occlusion or identity confusion. Overall, the contribution is limited and primarily experimental. I therefore recommend rejection of this paper in its current form.

**Questions:**

Major Concerns:
1. Frame Rate Comparison: FPS is compared with other state-of-the-art (SoTA) methods, but the results should also include performance on edge devices to demonstrate real-time feasibility.
2.  Outdated Baselines: Many compared SoTA methods are outdated. Recent trackers such as SMILETrack (AAAI 2024) and methods (MeMoTTR and MOTIP ) reported in CVPR 2023–2025 should be included for a fair comparison.
3. Efficiency Drop with ReID: As shown in Table 3, adding the ReID module significantly decreases efficiency, indicating scalability issues.
4.  Computation Overhead of MeKF: The MeKF component notably reduces performance from 74.5 FPS to 60.8 FPS, highlighting its inefficiency.
5.  Lack of Real-time Capability: The overall system is not real-time, especially when deployed on edge devices such as Jetson Nano or NX.
6.  Incomplete FPS Reporting: In Table 2, FPS results are missing for several compared methods. These should be provided for a complete and fair analysis or reported in another table.
7.  Missing Benchmark Evaluations: The paper does not evaluate on MOT17 or MOT20, which are standard benchmarks for MOT. This omission weakens the validity of the results and comparability with existing work.

**Details Of Ethics Concerns:**

No.

---

> ### Comment · Reviewer_DhRY · 2025-11-12
>
> Most concerns focus around the FPS. I shared some of my thoughts via the official comments. I hope the reviewer can take a look and re-evaluate the value of the work beyond runtime.

---

> ### Author Response · Authors · 2025-12-03
> **Response to Weakness 1.1: Computational Cost and Training Efficiency**
>
> **Note: References cited in this and subsequent responses are listed in the final comment.**
>
> * **Computational cost**
>   Actually, the computational cost of MeKF is not expensive. This is because integrating a lightweight LSTM (with a hidden dimension of only 32 units) into the Bayesian filtering framework is sufficient to achieve strong performance, as demonstrated in our sensitivity analysis (please see **Appendix E.2, L 1125-1133** for details).
>   Compared to trackers that employ purely neural network-based filters, such as DiffMOT (using Diffusion-based filter) [1] and TrackSSM (using Mamba-based filter) [2] for state prediction, the computational cost of MeKF is significantly lower. As the FPS results added in **Table 1** of the revised manuscript (**L 378-397**) show, our MeMoSORT achieves a higher FPS than both DiffMOT, TrackSSM, and transformer-based end-to-end trackers (MOTRv2, MeMOTR, MOTIP), **surpassing them at least 6.1 FPS**. We detail the specific quantitative comparisons below:
>
>   **Table 1**. Performance on DanceTrack test set, FPS on validation. The best/second results are shown in **bold**/$\underline{\mathrm{underlined}}$.
>   |Methods|HOTA↑|AssA↑|IDF1↑|DetA↑|MOTA↑|FPS↑|
>   |:-|:-:|:-:|:-:|:-:|:-:|:-:|
>   |**End-to-end tracker**|
>   |MOTRv2 [3]|69.9|64.4|76.0|83.7|92.1|10.2|
>   |MeMOTR [4]|63.4|52.3|65.5|77.0|85.4|17.7|
>   |MOTIP [5]|69.6|60.4|74.7|80.4|90.6|19.8|
>   |**KF-based filter**|
>   |ByteTrack [6]|47.7|32.1|53.9|71.0|89.6|**35.8**|
>   |OC-SORT [7]|55.1|40.4|54.9|80.4|92.2|-|
>   |Deep OC-SORT [8]|61.3|45.8|61.5|82.2|92.3|-|
>   |C-BIoU [9]|60.6|45.4|61.6|81.3|91.6|-|
>   |Hybrid-SORT [10]|65.7|-|67.4|-|91.8|15.5|
>   |TrackTrack [11]|$\underline{\text{66.5}}$|$\underline{\text{52.9}}$|$\underline{\text{67.8}}$|-|**93.6**|-|
>   |**Sliding window-based filter**|
>   |MotionTrack [12]|58.2|41.7|58.6|81.4|91.3|-|
>   |DiffMOT [1]|62.3|47.2|63.0|$\underline{\text{82.5}}$|92.8|22.7|
>   |**Implicit memory-based filter**|
>   |MambaMOT [13]|56.1|39.0|54.9|80.8|90.3|$\underline{\text{28.8}}$|
>   |TrackSSM [2]|57.7|41.0|57.5|81.5|92.2|20.3|
>   |DeepMove SORT [14]|63.0|48.6|65.0|82.0|92.6|-|
>   |**MeMoSORT** (Ours)|**67.9**|**54.3**|**68.0**|**85.0**|$\underline{\text{93.4}}$|$\underline{\text{28.8}}$|
>
> * **Training efficiency**
>   We clarify that MeKF avoids pure RNN instability by leveraging the Kalman filter as a physical prior. By tasking the LSTM to maintain temporal memory and generate only the residual terms (i.e., $\mathbf{\Delta}_t^\mathbb{F}$) rather than learning dynamics from scratch, this **residual learning paradigm provides a inductive bias** that simplifies the optimization process. Consequently, the model converges rapidly with standard optimizers, avoiding the complex tuning and multi-stage training typical of end-to-end Transformers.

---

> ### Author Response · Authors · 2025-12-03
> **Response to Weakness 1.2: Additional Benchmarks on MOT17 and MOT20**
>
> We have incorporated full evaluations on the MOT17 and MOT20 benchmarks into the revised manuscript (please refer to **Tables 7 and 8, L 1033-1082** for details). The experimental results demonstrate that our method maintains competitive performance on these challenging datasets, proving its robustness and broad applicability. The specific quantitative comparisons are presented in the tables below:
>
>   **Table 2.** Performance comparison with SOTA methods on the MOT17 test set. The best/second results are shown in **bold**/$\underline{\mathrm{underlined}}$.
>   |Methods|HOTA↑|AssA↑|IDF1↑|DetA↑|MOTA↑|IDs↓|FP($10^4$)↓|FN($10^4$)↓|
>   |:-|:-:|:-:|:-:|:-:|:-:|:-:|:-:|:-:|
>   |ByteTrack [6]|63.1|-|77.3|-|$\underline{\text{80.3}}$|2196|2.55|**8.37**|
>   |OC-SORT [7]|63.2|63.2|77.5|-|78.0|1950|**1.51**|10.80|
>   |C-BIoU [9]|**64.1**|63.7|**79.7**|**64.8**|**81.1**|-|-|-|
>   |BUSCA [15]|$\underline{\text{63.9}}$|64.2|79.2|$\underline{\text{63.9}}$|78.6|**1428**|2.46|$\underline{\text{9.45}}$|
>   |TOPICTrack [16]|$\underline{\text{63.9}}$|$\underline{\text{64.3}}$|78.6|63.7|78.8|$\underline{\text{1515}}$|$\underline{\text{1.70}}$|10.11|
>   |**MeMoSORT** (ours)|$\underline{\text{63.9}}$|**64.5**|$\underline{\text{79.3}}$|63.6|78.7|2058|2.02|9.77|
>
>   **Table 3.** Performance comparison with SOTA methods on the MOT20 test set. The best/second results are shown in **bold**/$\underline{\mathrm{underlined}}$.
>   |Methods|HOTA↑|AssA↑|IDF1↑|DetA↑|MOTA↑|IDs↓|FP($10^4$)↓|FN($10^4$)↓|
>   |:-|:-:|:-:|:-:|:-:|:-:|:-:|:-:|:-:|
>   |ByteTrack [6]|61.3|-|75.2|-|$\underline{\text{77.8}}$|1223|2.62|$\underline{\text{8.76}}$|
>   |OC-SORT [7]|62.1|62.0|75.9|-|75.5|$\underline{\text{913}}$|1.80|10.80|
>   |BPMTrack [17]|$\underline{\text{62.3}}$|60.9|$\underline{\text{76.7}}$|**63.9**|**78.3**|1314|2.86|**8.25**|
>   |BUSCA [15]|61.8|63.5|76.3|$\underline{\text{60.3}}$|72.7|1006|1.38|12.63|
>   |TOPICTrack [16]|**62.6**|**65.4**|**77.6**|60.0|72.4|**869**|**1.10**|13.11|
>   |**MeMoSORT** (ours)|61.9|$\underline{\text{63.8}}$|75.7|60.2|72.5|1200|$\underline{\text{1.26}}$|12.83|
>
>   As shown in **Tables 2** and **3**, MeMoSORT achieves results close to SOTA methods like C-BIoU and TOPICTrack. Notably, on MOT17, our method achieves the best AssA of 64.5, outperforming recent baselines like TOPICTrack and BUSCA, and we rank the second HOTA. The slight variance in performance can be attributed to the characteristics of the datasets:
>
>   * **Distinct motion patterns**: Pedestrians in MOT17/20 typically move in a linear and stable manner [2, 14], making standard KFs based on the Markovian assumption sufficient for motion prediction. Conversely, targets in DanceTrack and SportsMOT **exhibit long-term non-Markovian dynamics** (e.g., a dancer consistently spinning after a specific jump), a complexity that our MeKF is explicitly designed to handle. Consequently, the distinct advantage of our MeKF in modeling complex motion is not fully exploited on MOT17/20, resulting in performance marginally superior to standard KF-based methods.
>
>   * **Distinct occlusion characteristics**: The motion characteristics of targets during occlusion in MOT17/20 significantly differ from those in DanceTrack and SportsMOT. For example, targets in the latter datasets often interact with complex and rapid movements during occlusion, whereas occluded pedestrians in MOT17/20 typically remain in a slow-motion. When targets within the surveillance area move slowly, the dynamic adaptability of Mo-IoU to varying speeds yields only limited gains.
>
>   * **Robust Generalization**: Overall, despite these limiting factors on MOT17 and MOT20, MeMoSORT still outperforms several trackers that do not utilize neural networks. This demonstrates that MeKF and Mo-IoU effectively solve the complex MOT problem without overfitting to the simplified scenarios in traditional benchmarks.

---

> ### Author Response · Authors · 2025-12-03
> **Response to Weaknesses 2 & 3: Feature Robustness and Theoretical Novelty**
>
> * **Independence from Camera Calibration and Metric Height**
>    Actually, our method **DOES NOT require metric height or camera calibration**. We do not estimate the ''reliable'' real-world height (e.g., in meters) of a person. Our HIoU, like standard IoU, operates directly on the pixel coordinates of bounding boxes in uncalibrated video frames. Our SOTA results on the uncalibrated DanceTrack and SportsMOT datasets (**L 378-424**) were achieved without any camera calibration, demonstrating the method's applicability.
>    Specifically, the HIoU focuses on the overlap of heights between the predicted box and detection box, which can be defined as a 1D-IoU on the vertical axis. It measures the overlap consistency using the intersection height ($\dot{l}_{t-1}$) relative to the union of heights, as defined in Section 3.4, Paragraph 4 of the manuscript.
>    Therefore, even if two targets have similar absolute heights (e.g., $\hat{h}'\_t \approx \widetilde{h}\_t$), their HIoU will be low if their **vertical positions differ** (e.g., one target vertical position is higher in the frame than the other), as this would result in a small intersection height ($\dot{l}\_{t-1}$). This is precisely what allows it to distinguish targets during occlusion or interaction.
>
> * **Substantial Theoretical Contributions Beyond Experiments**
>    We respectfully clarify that our contributions are not solely experimental. Our proposed modules are grounded in theoretical and mathematical principles:
>    * The MeKF is based on a **theoretical derivation** as a neural network–aided Bayesian filter (detailed in **Section 3.3** and **Appendix B** **L 835-991**).
>    * The Mo-IoU has a **clear motivation** and a **precise mathematical definition** (detailed in **L 279-323**), rooted in geometric properties and motion statistics.

---

> ### Author Response · Authors · 2025-12-03
> **Response to Questions**
>
> 1. **Frame Rate Comparison**
>    We respectfully clarify that while the primary focus of this paper is on the **algorithmic design** and **theoretical performance comparison**, we did not ignore the importance of a lightweight network design.
>
>     As evidenced in the **Table 4**, our MeKF is significantly lighter than purely neural network-based filter (e.g., DiffMOT's D²MP [1]). Specifically, it achieves a highly favorable trade-off with minimal parameter count and reduced computational cost (FLOPs). This structural simplicity ensures that our model is not only theoretically principled **but also inherently suitable for real-time deployment on edge devices** using standard optimization pipelines.
>
>     **Table 4.** Computational cost comparison.
>     | Filter | Total params (M) ↓ | FLOPs (M) ↓ | FPS ↑    |
>     | :- | :--: | :--: | :--: |
>     | D²MP (DiffMOT) [1] | 6.53 | 28.9 | 22.7 |
>     | **MeKF (MeMoSORT)** | **0.0062** | **0.05** | **28.8** |
>
>     A hardware-specific deployment study is **beyond the scope** of this work. Moreover, in TBD frameworks, **the computational bottleneck lies in the detector**, as our lightweight MeKF adds negligible overhead in comparison.
>
> 2. **Outdated Baselines**
>    Following your recommendation to ensure a comprehensive and up-to-date comparison, we have updated our experiments to include representative SOTA methods from 2023 to 2025. Specifically, we added comparison to TrackTrack (_CVPR 2025_), MOTIP (_CVPR 2025_), MeMOTR (_CVPR 2023_), and MOTRv2 (_CVPR 2023_) on DanceTrack test set, as well as TOPICTrack (_TIP 2025_), BUSCA (_ECCV 2024_), and BPMTrack (_TIP 2024_) on MOT17 and MOT20 benchmarks.
>
>    For the specific results, please refer to **Tables 1, 2 and 3** presented in our responses to **Weakness 1.1 and Weakness 1.2**. For full details and analysis, please refer to the revised manuscript **Table 1, L 378-397** and **Tables 7 and 8, L 1041-1060**.
>
> 3. **Efficiency Drop with ReID**
>   This significant degradation is an **inevitable consequence of incorporating such computationally intensive modules**. For instance, SMILETrack (recommended by the reviewer) also exhibits a significant FPS drop when its ReID module is utilized, as the FPS is **37.5** without ReID but drops to **5.6** ($\mathbf{\downarrow 85.1\%}$) with ReID. This is a common trade-off, and high-efficiency trackers face the same issue. Similarly, ByteTrack drops from **29.6** to **11.8** FPS ($\mathbf{\downarrow 60.1\%}$) when ReID is added, and Hybrid-SORT drops from **27.8** to **15.5** FPS ($\mathbf{\downarrow 44.2\%}$).
>
>     Notably, our ablation study demonstrates that **MeMoSORT does not strictly rely on ReID**. As shown in the **Table 5** below (please refer to **Table 3, L 448-448** in the revised manuscript for details), driven by the robust tracking capabilities of MeKF and Mo-IoU, removing ReID results in a negligible HOTA drop of only 0.37 (77.91 → 77.54):
>
>    **Table 5.** Selected ablation results of MeMoSORT's key components on the DanceTrack validation set.
>
>    |MeKF|Mo-IoU|ReID|HOTA↑|AssA↑|IDF1↑|MOTA↑|DetA↑|FPS↑|
>    |:-:|:-:|:-:|:-:|:-:|:-:|:-:|:-:|:-:|
>    |✓|✓|-|77.54|64.73|76.92|**97.74**|92.93|**49.4**|
>    |✓|✓|✓|**77.91**|**65.21**|**77.49**|97.73|**93.13**|28.8|
>
> 4. **Computation Overhead of MeKF**
>    It is true that MeKF's use of neural network components (i.e., LSTM) increases computational overhead. Compared to the traditional KF-based tracker ByteTrack (which has no NN components in state prediction), this overhead is justified by **a significant improvement in accuracy**, as the results in **Table 3, L 448-448** of the revised manuscript show. MeMoSORT with only MeKF achieves a HOTA of **67.41**, outperforming ByteTrack's 56.94 by a substantial margin of **+10.47**, while also boosting AssA and IDF1 by **+14.66** and **+18.23**, respectively. This is all achieved while still maintaining a competitive FPS of 60.8.
>
>    Moreover, we have added a comparison of both performance and FPS against end-to-end transformer-based trackers, such as MOTRv2 [3], MeMOTR [4], and MOTIP [5]. Notably, ours MeMoSORT achieves **28.8 FPS**, surpassing these methods by significant margins of **+18.6, +11.1, and +9.0 FPS**, respectively. Please refer to **Table 1** in our **Response to Weakness 1.1: Computational Cost and Training Efficiency**. The results demonstrate that our method achieves comparable tracking performance while requiring **significantly lower computational cost** than these heavy architectures.

---

> ### Author Response · Authors · 2025-12-03
> **Response to Questions**
>
> 5. **Lack of Real-time Capability**
>    We appreciate the reviewer's comment. We would like to reiterate that the core contribution of this work is centered on **algorithmic design** and its **theoretical performance**, rather than on system-level deployment. Nevertheless, we designed our method in an efficiency manner. It is computationally lightweight, possessing a small parameter footprint and a simple design, which makes it highly amenable to future optimization for resource-constrained platforms.
>
>    Notably, in the TBD paradigm, the **computational bottleneck lies in the detector**. Since our MeKF adds negligible latency compared to the baselines, it follows logically that if those compared methods are feasible for edge deployment, ours is inherently even more suitable.
>
>    While we agree that a full deployment study on edge devices is an important engineering task, this hardware-specific optimization is **outside the defined scope** of our current paper.
>
> 6. **Incomplete FPS Reporting**
>    To ensure a complete and fair comparison, we have updated **Table 1** in the revised manuscript (**L 378-398**) to include the FPS results on the DanceTrack validation set. Furthermore, we have added comparative experiments against recent methods such as MOTRv2 [3], MeMOTR [4], and MOTIP [5]. The key results regarding inference speed are presented below:
>    **Table 6.** Inference speed comparison (FPS) on the DanceTrack validation set.
>    | Methods      | **MeMoSORT** | MOTRv2 | MeMOTR | MOTIP | ByteTrack | Hybrid-SORT | DiffMOT | MambaMOT | Track SSM |
>    |--------------|--------------|--------|--------|-------|-----------|-------------|---------|----------|-----------|
>    | FPS ↑        | $\underline{\text{28.8}}$ | 10.2   | 17.7   | 19.8  | **35.8**  | 15.5        | 22.7    | $\underline{\text{28.8}}$ | 20.3      |
>
>
> 7. **Missing Benchmark Evaluations**
>    We have incorporated full evaluations on the MOT17 and MOT20 benchmarks, please refer to **Tables 2 and 3** in our **Response to Weakness 1.2: Additional Benchmarks on MOT17 and MOT20**. For the specific results, please see **Tables 7 and 8, L 1041-1060** of the revised manuscript for details.

---

> ### Author Response · Authors · 2025-12-03
> **References**
>
> [1] W. Lv, et al. "Diffmot: A real-time diffusion-based multiple object tracker with non-linear prediction." (_CVPR 2024_)
> [2] B. Hu, et al. "Trackssm: A general motion predictor by state-space model." arXiv preprint arXiv:2409.00487, 2024.
> [3] Y. Zhang, et al. "Motrv2: Bootstrapping end-to-end multi-object tracking by pretrained object detectors." (_CVPR 2023_)
> [4] R. Gao, et al. "MeMOTR: Long-term memory-augmented transformer for multi-object tracking." (_CVPR 2023_)
> [5] R. Gao, et al. "Multiple object tracking as id prediction." (_CVPR 2025_)
> [6] Y. Zhang, et al. "Bytetrack: Multi-object tracking by associating every detection box." (_ECCV 2022_)
> [7] J. Cao, et al. "Observation-centric sort: Rethinking sort for robust multi-object tracking." (_CVPR 2023_)
> [8] G. Maggiolino, et al. "Deep oc-sort: Multi-pedestrian tracking by adaptive re-identification." (_ICIP 2023_)
> [9] F. Yang, et al. "Hard to track objects with irregular motions and similar appearances? make it easier by buffering the matching space." (_WACV 2023_)
> [10] M. Yang, et al. "Hybrid-sort: Weak cues matter for online multi-object tracking." (_AAAI 2024_)
> [11] K. Shim, et al. "Focusing on Tracks for Online Multi-Object Tracking." (_CVPR 2025_)
> [12] Z. Qin, et al. "Motiontrack: Learning robust short-term and long-term motions for multi-object tracking." (_CVPR 2023_)
> [13] H.-W. Huang, et al. "MambaMOT: State-Space Model as Motion Predictor for Multi-Object Tracking." (_ICASSP 2025_)
> [14] M. Adžemović, et al. "Engineering an Efficient Object Tracker for Non-Linear Motion." arXiv preprint arXiv:2407.00738 (2024).
> [15] L. Vaquero, et al. "Lost and found: Overcoming detector failures in online multi-object tracking." (_ECCV 2024_)
> [16] X. Cao, et al. "TOPIC: a parallel association paradigm for multi-object tracking under complex motions and diverse scenes." (_IEEE TIP 2025_)
> [17] Y. Gao, et al. "BPMTrack: Multi-object tracking with detection box application pattern mining." (_IEEE TIP 2024_)

---

### Comment · Reviewer_DhRY · 2025-11-12
**Official comment from Reviewer DhRY**

I have read through the feedbacks from the other two reviewers. Here is something I want to raise the discussion.

Most reviewers mentioned the paper weakness in the runtime, and I would like to raise a general concern regarding the evaluation focus in this discussion. While computational efficiency and runtime (FPS) are certainly important factors, I believe the current round of reviews places excessive emphasis on these aspects (especially reviewer uXas, 5 out of 7 of his reviews are concerns regarding the runtime), in some cases even suggesting evaluations on edge devices such as Jetson Nano or NX. Such requirements go beyond the reasonable expectation for a research paper introducing a new method during the rebuttal stage. **To my best knowledge, I never seen any top-conference MOT papers run FPS evaluation on edge device.**

When developing new algorithms, there is often an inherent trade-off between accuracy and efficiency. Novel methods typically prioritize demonstrating conceptual and empirical advances, while efficiency optimizations can follow in subsequent work. Therefore, I encourage reviewers to evaluate the paper more holistically by considering the paper's novelty, insight, and contribution to advancing the field of MOT, rather than focusing disproportionately on the method's FPS. **If everyone sets a constraint on the FPS of a new paper submission in the MOT field, there will be no new papers after ByteTrack.**

I respectfully suggest that reviewers reconsider their final ratings with this broader perspective in mind, ensuring that the evaluation reflects the paper’s scientific value beyond raw FPS comparisons.

---

### Meta-Review · Area_Chair_1kYR · 2025-12-27

**Summary:**

The main concerns raised by reviewers initially include (a) limited experimental coverage (missing MOT17/MOT20 and recent baselines), (b) doubts about the novelty and principled nature of MeKF and Mo-IoU, (c) computational efficiency and real-time feasibility, and (d) whether the design choices were ad-hoc or well-justified. One reviewer expressed strong skepticism and recommended rejection, while another provided a marginally positive assessment, and a third reviewer was strongly positive, emphasizing the method’s effectiveness in complex human-motion scenarios. While the revised results strengthen validity, the gains appear concentrated in a narrow class of human-motion datasets and do not translate into a clearly compelling and broadly impactful advance over established MOT baselines.

**Reviewer Concerns:**

Concerns addressed:
1. Missing benchmarks and outdated baselines. The authors added evaluations on MOT17 and MOT20 and comparisons with recent trackers, resolving the primary concern about insufficient experimental validation.

2. Limited metrics. Additional metrics were reported, and the choice of HOTA as the primary metric was well justified.

3. Computational cost and efficiency. Detailed FPS, FLOPs, and parameter analyses were added, showing competitive performance in the accuracy-speed trade-off.

4. Hyperparameter Sensitivity. Added sensitivity studies for Mo-IoU/MAT parameters and some detector-dependency analysis for MeKF training.

4. Generalization and detector dependence. Cross-dataset experiments and detector-quality sensitivity studies addressed concerns about domain shift and reliance on strong detectors.


Concerns unaddressed:
1. Novelty and significance. Even with added motivation and sensitivity analysis, Mo-IoU/MAT and the MeKF hybridization still read as heuristic add-ons rather than a clearly principled and general contribution.

2. Strength vs. scope. Improvements appear most obvious on a narrow set of human-motion datasets; results on standard benchmarks are closer to existing methods, weakening the claim of a compelling advance.

3. Practicality claims. The rebuttal argues real-time feasibility and “lightweight” design, but system-level evidence (especially for constrained/edge settings raised by reviewers) remains limited, leaving practicality claims less convincing.

**Reviewer Scores:**

Reviewer uXas would still likely remain below acceptance due to novelty/practicality skepticism.

Reviewer iKTz is likely to stay around borderline, and should be weighted cautiously given the reviewer’s statement of low confidence.

Reviewer DhRY would likely stay ar borderline accept, since the rebuttal strengthens points already highlighted in the review.

Overall, the paper is solid and the rebuttal addresses many of the reviewer concerns, but the contributions and evidence of broad impact are not strong enough to meet the bar. Overall, it is a reasonable piece of work, but it falls just short of acceptance.

---

### Decision · Program_Chairs · 2026-01-26

Reject